# FoxA1 and FoxA2 drive gastric differentiation and suppress squamous identity in NKX2-1-negative lung cancer

Soledad A Camolotto[1], Shrivatsav Pattabiraman[1†], Timothy L Mosbruger[2†], Alex Jones[1], Veronika K Belova[1], Grace Orstad[1], Mitchell Streiff[1], Lydia Salmond[1], Chris Stubben[2], Klaus H Kaestner[3], Eric L Snyder[1]*

[1]Department of Pathology and Huntsman Cancer Institute, University of Utah, Salt Lake City, United States; [2]Bioinformatics Shared Resource, Huntsman Cancer Institute, University of Utah, Salt Lake City, United States; [3]Department of Genetics and Institute for Diabetes, Obesity, and Metabolism, Perelman School of Medicine, University of Pennsylvania, Pennsylvania, United States

*For correspondence:
eric.snyder@hci.utah.edu

†These authors contributed equally to this work

Competing interests: The authors declare that no competing interests exist.

**Abstract** Changes in cancer cell identity can alter malignant potential and therapeutic response. Loss of the pulmonary lineage specifier NKX2-1 augments the growth of KRAS-driven lung adenocarcinoma and causes pulmonary to gastric transdifferentiation. Here, we show that the transcription factors FoxA1 and FoxA2 are required for initiation of mucinous NKX2-1-negative lung adenocarcinomas in the mouse and for activation of their gastric differentiation program. *Foxa1/2* deletion severely impairs tumor initiation and causes a proximal shift in cellular identity, yielding tumors expressing markers of the squamocolumnar junction of the gastrointestinal tract. In contrast, we observe downregulation of FoxA1/2 expression in the squamous component of both murine and human lung adenosquamous carcinoma. Using sequential in vivo recombination, we find that FoxA1/2 loss in established KRAS-driven neoplasia originating from SPC-positive alveolar cells induces keratinizing squamous cell carcinomas. Thus, NKX2-1, FoxA1 and FoxA2 coordinately regulate the growth and identity of lung cancer in a context-specific manner.
DOI: https://doi.org/10.7554/eLife.38579.001

## Introduction

Cancer progression is often accompanied by profound changes in cellular identity. Cellular identity, or differentiation state, influences not only intrinsic malignant potential, but also response to therapy, even in tumors harboring the same targetable mutations (*Cohen and Settleman, 2014*). Although tissue of origin is a major determinant of cancer cell identity, cancer cells can also undergo lineage switching in the course of their natural history and in response to the selective pressure of targeted therapy. In lung adenocarcinoma, absence of the pulmonary lineage specifier NKX2-1/TTF1 correlates with non-pulmonary cellular identities and poor prognosis compared with NKX2-1-positive tumors (*Barletta et al., 2009*; *Cardnell et al., 2015*). Moreover, lung adenocarcinomas can undergo lineage switching during the evolution of drug resistance that reduces their dependence on the oncogenic signaling pathway being targeted (*Rotow and Bivona, 2017*). Taken together, these observations indicate that there is a need to understand the critical regulators of cancer cell identity.

In previous work, we and others have shown that loss of NKX2-1 is sufficient to cause lineage switching in a mouse model of KRAS[G12D]-driven lung adenocarcinoma (*Maeda et al., 2011*; *Snyder et al., 2013*; *Tata et al., 2018*). *Nkx2-1* deletion in established tumors causes cancer cells to shed their pulmonary identity and adopt a gastric-like differentiation state characterized by extensive mucin production and expression of multiple gastrointestinal markers, including HNF4α and

**eLife digest** Among all cancers, lung cancers cause the most deaths worldwide. There are many different types of lung cancer, each of which contain lung cancer cells that look different. As a general rule, lung cancer cells that look the most like healthy lung cells are the least aggressive. Cancer cells that take on the appearance of other tissues in the body are more aggressive and often respond poorly to treatment. In one uncommon type of lung cancer called invasive mucinous adenocarcinoma (IMA, for short), the cancer cells start to resemble the cells that line the inside of the stomach. For example, these lung cancer cells activate genes more typically active in stomach cells, and they start to make a lot of mucus.

Previous studies with mice showed that losing a single protein called NKX2-1 can cause this switch from lung to stomach cell identity. However, it is not clear exactly how this switch happens and which other proteins are involved. Camolotto et al. have now addressed these issues by studying two DNA-binding proteins called FoxA1 and FoxA2. There were two main reasons for choosing these specific proteins. First, they can physically interact with the NKX2-1 protein, so losing NKX2-1 affects how FoxA1 and FoxA2 interact with DNA. Second, the two proteins switch on many of the stomach-related genes that are also activated in IMA.

Camolotto et al. activated a gene that commonly drives lung cancer and deleted the gene for NKX2-1 in the lungs of mice, mimicking IMA. As expected, these mice developed lung tumors that resembled stomach tissue. When the genes for FoxA1 and FoxA2 were deleted at the same time, the tumors stopped producing the mucus-related proteins. Further experiments showed that these cancer cells adopt a different cell identity also found in the digestive tract. Mice with tumors lacking both FoxA1 and FoxA2 survived for longer than those still containing these proteins. Lastly, when the genes for NKX2-1, FoxA1 and FoxA2 were deleted later, in lung tumors that had already formed, the outcome was a more aggressive type of lung cancer that also occurs in human patients.

These experiments demonstrate that losing FoxA1 and FoxA2 at different times affects what kind of lung tumor can grow. Future studies will need to examine how these different lung cancer types respond to therapy and whether lung cancer cells switch identities to evade therapy. This knowledge may eventually lead to new treatments for lung cancer patients.
DOI: https://doi.org/10.7554/eLife.38579.002

Gastrokine 1. These tumors morphologically resemble a subtype of human lung cancer called invasive mucinous adenocarcinoma (IMA), which also expresses gastrointestinal markers and is predominantly driven by *KRAS* mutations (*Guo et al., 2017*). Approximately 10–15% of human lung adenocarcinomas express HNF4α with no detectable NKX2-1 (9), including both IMAs and more moderately differentiated tumors. In many of these tumors, the *NKX2-1* gene appears to be silenced by genetic and/or epigenetic mechanisms (*Hwang et al., 2016*; *Matsubara et al., 2017*). Aside from NKX2-1 itself, the Polycomb Repressive Complex 2 (PRC2) appears to play a role in suppressing mucinous differentiation in KRAS-driven, p53-deficient lung adenocarcinoma (*Serresi et al., 2016*). However, the precise mechanisms by which a gastric gene expression program is activated in NKX2-1-deficient tumors remain to be fully elucidated.

Many of the gastrointestinal transcripts expressed in IMA are known targets of the forkhead box transcription factors FoxA1 and FoxA2 (FoxA1/2). These transcription factors govern the development of a variety of tissues and are expressed in both the adult lung and GI tract (reviewed in *Golson and Kaestner, 2016*). FoxA1/2 are also expressed in both murine and human IMA (*Figure 1A* and *Figure 1—figure supplement 1A–B*). We previously found that *Nkx2-1* deletion in autochthonous lung tumors caused FoxA1/2 to re-localize from the regulatory elements of pulmonary-specific genes (such as *Sftpa1*) to those of genes (such as *Hnf4a*) that are expressed in both the GI tract and IMA (*Snyder et al., 2013*). Given that NKX2-1 physically interacts with FoxA1/2 (*Snyder et al., 2013*; *Minoo et al., 2007*), we hypothesized that NKX2-1 promotes FoxA1/2 interaction with regulatory elements of the pulmonary differentiation program at the expense of those governing gastric identity. However, these data did not demonstrate a functional role for FoxA1/2 in the activation of the gastric program in these tumors. To address this question directly, we used conditional alleles of *Foxa1* (*Gao et al., 2008*) and *Foxa2* (*Sund et al., 2000*) to abrogate their function in

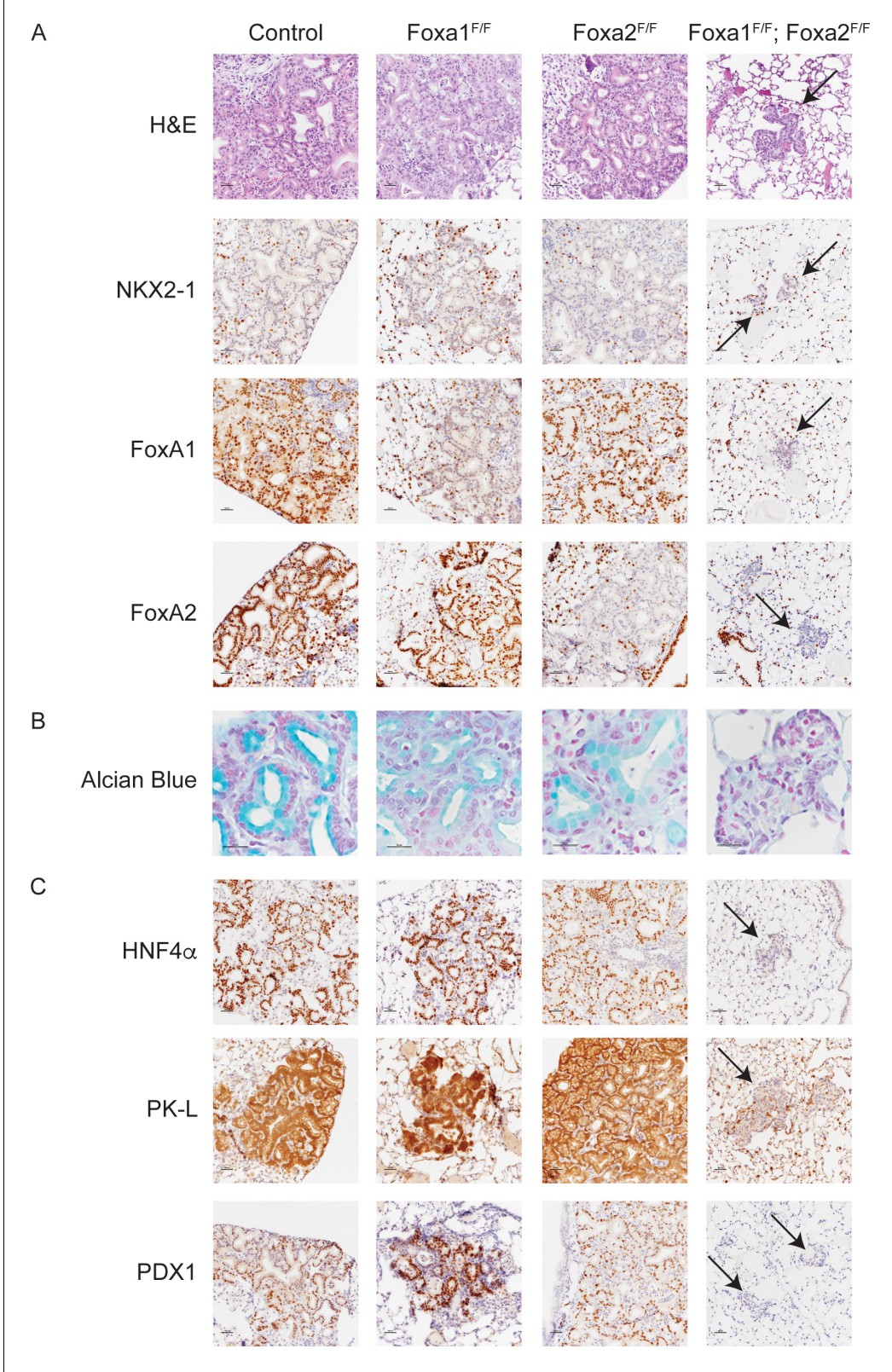

**Figure 1.** FoxA1 and FoxA2 are required for mucinous lung adenocarcinoma formation. Photomicrographs of lung neoplasia arising 11 weeks after initiation with PGK-Cre lentivirus. All mice are $Kras^{LSL-G12D/+}$; $Nkx2-1^{F/F}$ and harbor conditional alleles of $Foxa1$ and/or $Foxa2$ as indicated. (**A**) Hematoxylin and eosin (H and E) and immunohistochemistry (IHC) for NKX2-1, FoxA1 and FoxA2. Arrows indicate neoplasia lacking expression of all

*Figure 1 continued on next page*

*Figure 1 continued*

three proteins. Scale bar: 100 microns. (B) Alcian blue stain for mucin production. Scale bar: 50 microns. (C) IHC for markers of gastrointestinal differentiation HNF4α, PK-L and PDX1. Scale bar: 100 microns.

DOI: https://doi.org/10.7554/eLife.38579.003

The following figure supplement is available for figure 1:

**Figure supplement 1.** FoxA1 and FoxA2 are required for mucinous lung adenocarcinoma formation.

DOI: https://doi.org/10.7554/eLife.38579.004

an autochthonous mouse model of NKX2-1-negative lung adenocarcinoma. We found that FoxA1/2 are critical and redundant regulators of both the gastric differentiation program and growth of NKX2-1-negative tumors. Moreover, we found that the cellular identity adopted by tumors was highly dependent on the context in which FoxA1/2 activity is lost, suggesting that a cell's baseline epigenetic state can influence the identity it adopts in response to changes in lineage specifier expression.

## Results

### FoxA1 and FoxA2 are required for development of invasive mucinous adenocarcinoma of the lung

To test the hypothesis that FoxA1/2 are required for lung adenocarcinoma cells to undergo a pulmonary to gastric lineage switch upon loss of NKX2-1 expression, we incorporated conditional alleles of *Foxa1* and *Foxa2* into a mouse model of NKX2-1-deficient lung adenocarcinoma (*Snyder et al., 2013*). In this model, intratracheal delivery of virus expressing Cre recombinase simultaneously activates a conditional allele of oncogenic *Kras* (*Kras^{LSL-G12D/+}*) and silences conditional alleles of *Nkx2-1* (*Nkx2-1^{F/F}*) alone or in addition to *Foxa1* (*Foxa1^{F/F}*) and/or *Foxa2* (*Foxa2^{F/F}*). Initial evaluation by morphology (H and E) and immunohistochemistry (IHC) showed that tumors lacking either FoxA1 or FoxA2 were indistinguishable from control tumors (*Figure 1A*). In sharp contrast, concomitant deletion of *Foxa1* and *Foxa2* led to the emergence of small neoplastic lesions (*Figure 1A*, right column) in the alveoli that were completely devoid of the glandular architecture and mucin production that characterizes NKX2-1-deficient tumors. Absence of mucin production was apparent by H and E staining and further demonstrated by Alcian Blue staining (*Figure 1B*) and IHC for Muc5AC (*Figure 1—figure supplement 1C*).

Given the dramatic change in the morphology of lung neoplasia lacking NKX2-1, FoxA1 and FoxA2, we used IHC to assess the differentiation state of these lesions. Cytokeratin 8 (CK8) was expressed in lesions arising in mice of all genotypes (*Figure 1—figure supplement 1C*), showing that cells lacking all three transcription factors retained an epithelial identity and did not undergo a complete epithelial to mesenchymal transition. HNF4α and PDX1 are transcription factors that regulate gastrointestinal differentiation and are expressed in human invasive mucinous adenocarcinoma and mouse models of this disease (*Snyder et al., 2013*; *Skoulidis et al., 2015*). Both transcription factors, as well as the HNF4α target PK-L, were undetectable in FoxA1/2-deficient neoplasia (*Figure 1C*). Additional markers of gastrointestinal differentiation, including Gastrokine 1, Cathepsin E and Galectin 4, were also not expressed in these lesions (*Figure 1—figure supplement 1C*). All these markers were retained in lesions lacking either FoxA1 or FoxA2 alone (*Figure 1C* and *Figure 1—figure supplement 1C*). Taken together, these data show that FoxA1 and FoxA2 are required for mucin production and key elements of the gastrointestinal differentiation program in NKX2-1-negative lung tumors in a functionally redundant manner.

### FoxA1 and FoxA2 are required at initiation for growth and proliferation of NKX2-1-negative lung adenocarcinoma

Most lesions in *Kras^{LSL-G12D/+}*; *Nkx2-1^{F/F}*; *Foxa1^{F/F}*; *Foxa2^{F/F}* mice exhibited complete loss of FoxA1/2 expression when analyzed at 11 weeks post-infection. However, these mice also harbored a variable but substantial quantity of tumors ('incomplete recombinants') that retained FoxA1 or FoxA2 as well as targets such as HNF4α (*Figure 2—figure supplement 1A–B*). Since incomplete recombinants were often larger than the lesions lacking NKX2-1, FoxA1 and FoxA2 (i.e. 'complete recombinants')

(*Figure 2—figure supplement 1B*), we speculated that they might have gradually outgrown the complete recombinants over time. Consistent with this possibility, we found that 5 weeks after tumor initiation, incomplete recombinants comprised a much smaller proportion of overall tumor burden than at 11 weeks (*Figure 2—figure supplement 1A–B*).

Based on these data, we chose the 5-week timepoint to quantitate tumor burden and proliferation rates among the different genotypes. We found that concomitant deletion of both *Foxa1* and *Foxa2* led to an approximately 10-fold reduction in tumor burden when measured at 5 weeks post-initiation (*Figure 2A*). This was accompanied by reduced lesion size and, to a lesser extent, fewer lesions/mm$^2$ (*Figure 2—figure supplement 1C–D*). In contrast, deletion of either *Foxa1* or *Foxa2* alone had little to no effect on tumor burden.

To determine why loss of FoxA1/2 activity caused such a severe inhibition of tumorigenesis, we analyzed proliferation and apoptosis in tumors of each genotype. BrdU incorporation was reduced by ~50% in FoxA1/2-deficient lesions in comparison with control lesions (*Figure 2B–C*). IHC for the

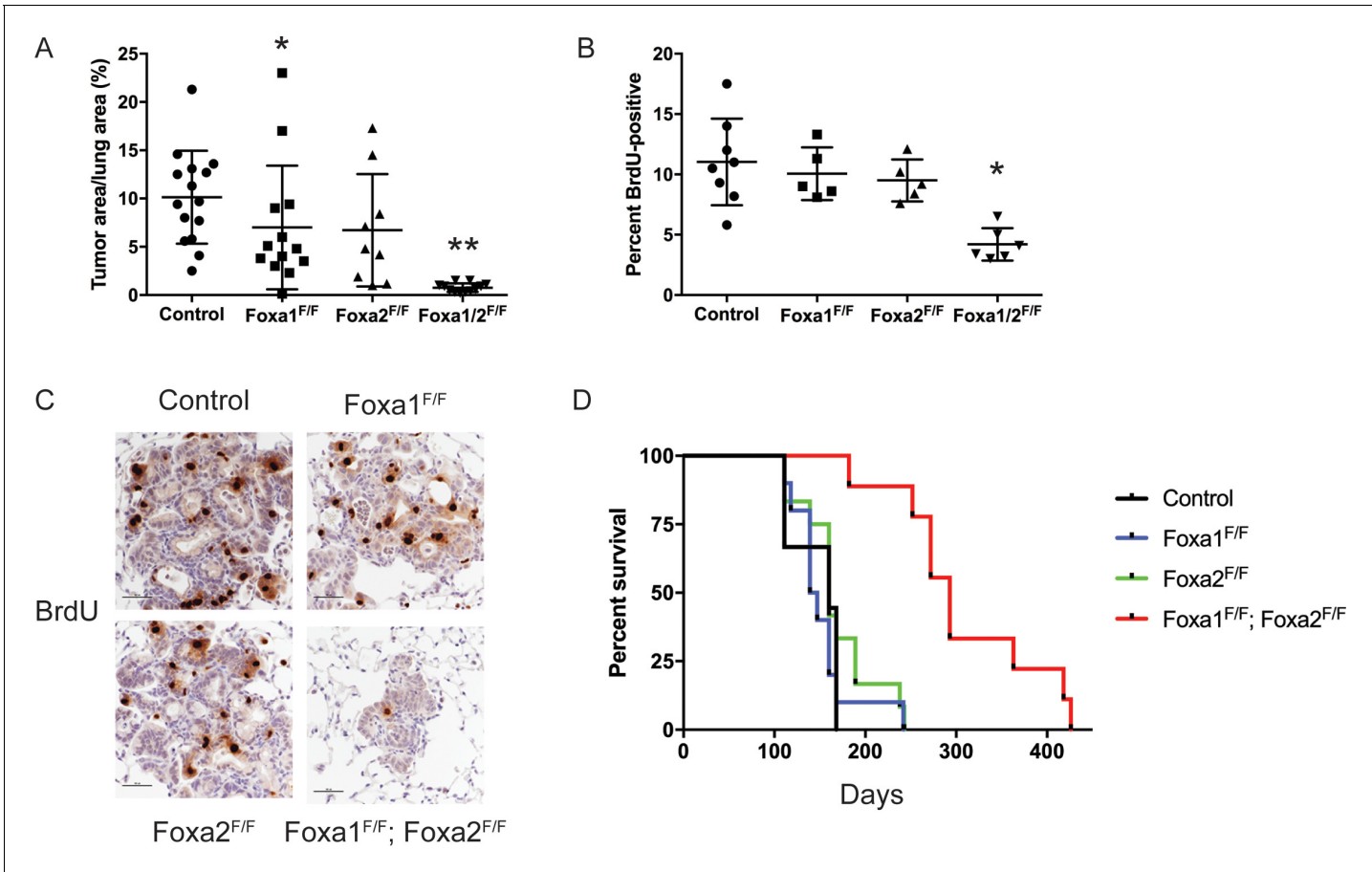

**Figure 2.** FoxA1 and FoxA2 are required for initiation and proliferation of NKX2-1-deficient lung adenocarcinoma. (**A**) Quantitation of tumor burden 5 weeks after initiation with PGK-Cre lentivirus in *Kras*[LSL-G12D/+]; *Nkx2-1*[F/F] mice of indicated genotype: control (n = 15), *Foxa1*[F/F] (n = 13), *Foxa2*[F/F] (n = 9) and *Foxa1*[F/F]; *Foxa2*[F/F] (n = 12). **p < 0.0002 vs. each group, Mann-Whitney. *p=0.0425 vs. control. Graphs represent mean ±S.D. (**B**) Quantitation of BrdU incorporation in lung neoplasia 5 weeks after initiation with PGK-Cre lentivirus in *Kras*[LSL-G12D/+]; *Nkx2-1*[F/F] mice of indicated genotype: control (n = 7), *Foxa1*[F/F] (n = 4), *Foxa2*[F/F] (n = 4) and *Foxa1*[F/F]; *Foxa2*[F/F] (n = 7). *p < 0.005 vs. each control, Mann-Whitney. Graphs represent mean ±S.D. (**C**) Representative IHC for BrdU in *Kras*[LSL-G12D/+]; *Nkx2-1*[F/F] mice of indicated genotype quantitated in *Figure 2B*. Scale bar: 100 microns. (**D**) Long-term survival after tumor initiation with PGK-Cre lentivirus in *Kras*[LSL-G12D/+]; *Nkx2-1*[F/F] mice of indicated genotype: control (n = 9), *Foxa1*[F/F] (n = 10), *Foxa2*[F/F] (n = 12) and *Foxa1*[F/F]; *Foxa2*[F/F] (n = 9). p < 0.0001, *Kras*[LSL-G12D/+]; *Nkx2-1*[F/F]; *Foxa1*[F/F]; *Foxa2*[F/F] mice vs. each control, Log-rank test.

DOI: https://doi.org/10.7554/eLife.38579.005

The following figure supplement is available for figure 2:

**Figure supplement 1.** FoxA1 and FoxA2 are required for initiation and proliferation of NKX2-1-deficient lung adenocarcinoma.
DOI: https://doi.org/10.7554/eLife.38579.006

proliferation markers MCM2 and KI67 also demonstrated that FoxA1/2-deficient lesions proliferate at a significantly lower rate than controls (*Figure 2—figure supplement 1E–G*). In contrast, the apoptotic rate of FoxA1/2-deficient lesions was no different than controls as measured by IHC for cleaved caspase-3 (*Figure 2—figure supplement 1H*).

In addition to these short-term measurements, we assessed the long-term impact of *Foxa1/2* deletion in a survival analysis (*Figure 2D*). Mice in the three control groups survived for a similar duration after tumor initiation (median survival 143–160 days). In contrast, deletion of both *Foxa1* and *Foxa2* led to a dramatic increase in survival (median survival 293 days). Histopathologic analysis showed that approximately 80% of the tumor burden in $Kras^{LSL-G12D/+}$; $Nkx2-1^{F/F}$; $Foxa1^{F/F}$; $Foxa2^{F/F}$ mice consisted of mucinous HNF4α-positive adenocarcinomas (*Figure 2—figure supplement 1A*). This suggests that these mice ultimately succumbed to growth of incomplete recombinants and that the complete recombinants likely had little impact on overall survival. We also noted extensive extracellular mucin secretion in the tumors of $Kras^{LSL-G12D/+}$; $Nkx2-1^{F/F}$; $Foxa1^{F/F}$ mice (*Figure 2—figure supplement 1I*). This phenomenon was rarely observed in tumors from other control groups, which predominantly produced intracellular mucin, suggesting that FoxA1 and FoxA2 likely have some specific functions in the regulation of the differentiation state of NKX2-1-negative adenocarcinoma. Taken together, these data show that lack of FoxA1/2 activity at tumor initiation severely impairs the proliferation and long-term growth potential of NKX2-1-negative lung adenocarcinoma.

## FoxA1 and FoxA2 are required for global activation of the gastric differentiation program in NKX2-1-negative lung adenocarcinoma

We next sought to analyze the changes in gene expression induced by deletion of *Foxa1* and *Foxa2* in NKX2-1-deficient tumors. Our mice harbor a Cre-dependent tdTomato reporter allele (*Madisen et al., 2010*) that enables tumor cell isolation by fluorescence-activated cell sorting (FACS). For sorting experiments, we initiated tumors with the Ad5-SPC-Cre adenovirus (*Sutherland et al., 2011*), which restricts Cre activity to SPC-positive lung epithelial cells, obviating the need to exclude stromal cells from the sorted population. (SPC-Cre induces lesions identical to lentiviral-driven Cre (Figure 6 and data not shown)). However, we lacked a cell surface marker that would enable us to differentially isolate complete from incomplete recombinants in $Kras^{LSL-G12D/+}$; $Nkx2-1^{F/F}$; $Foxa1^{F/F}$; $Foxa2^{F/F}$ mice during sorting.

Single-cell RNA-Seq can be used to deconvolute gene expression profiles of mixed cell populations from the murine lung bioinformatically and thereby assign an identity to each cell (*Treutlein et al., 2014*). We therefore proceeded with single-cell RNA-Seq analysis on FACS-sorted lung tumor cells via the Fluidigm C1 Autoprep microfluidic system. We sorted tumor cells from one $Kras^{LSL-G12D/+}$ mouse, one $Kras^{LSL-G12D/+}$; $Nkx2-1^{F/F}$ mouse, and two $Kras^{LSL-G12D/+}$; $Nkx2-1^{F/F}$; $Foxa1^{F/F}$; $Foxa2^{F/F}$ mice. After Illumina sequencing and transcript quantitation, we used the SC3 clustering package (*Kiselev et al., 2017*) for quality control, filtering and clustering. A total of 134 cells were considered to be of sufficient quality for further analysis (*Supplementary files 1–2*), which yielded three distinct clusters (tSNE plot, *Figure 3A*).

Cluster 1 (C1, n = 62 cells) contained cells from mice of all three genotypes. Using the SC3 package, we identified marker genes for this cluster (defined as 'genes that are highly expressed in only one of the clusters and are able to distinguish one cluster from all the remaining ones'). These included canonical NKX2-1 target genes *Sftpa1* and *Sftpb* (*Supplementary file 3*). From these data, we infer that C1 represents tumor cells that are phenotypically NKX2-1-positive. In contrast, cluster 2 (C2, n = 31 cells) only contained cells from $Kras^{LSL-G12D/+}$; $Nkx2-1^{F/F}$ and $Kras^{LSL-G12D/+}$; $Nkx2-1^{F/F}$; $Foxa1^{F/F}$; $Foxa2^{F/F}$ mice. Numerous gastrointestinal transcripts were identified as marker genes for this cluster, including *Hnf4a*, *Gkn1*, *Lgals4* and *Ctse*. Thus, C2 appears to include incomplete recombinants from $Kras^{LSL-G12D/+}$; $Nkx2-1^{F/F}$; $Foxa1^{F/F}$; $Foxa2^{F/F}$ mice that express sufficient levels of FoxA1 and/or FoxA2 to maintain a gastric differentiation state. In contrast, cluster 3 (C3, n = 41 cells) contained only cells from $Kras^{LSL-G12D/+}$; $Nkx2-1^{F/F}$; $Foxa1^{F/F}$; $Foxa2^{F/F}$ mice and expressed marker genes not characteristic of either a pulmonary or gastric differentiation state, suggesting that C3 likely contains cells completely deficient for NKX2-1, FoxA1 and FoxA2 (i.e. complete recombinants).

Several different analyses further validated our classification of C1 and C2 as NKX2-1-positive and NKX2-1-negative cells, respectively. First, we identified differentially expressed genes between C1 and C2 using an independent software package (SCDE) and found that many pulmonary and gastric transcripts were differentially expressed between the two clusters (*Supplementary file 3*). We then

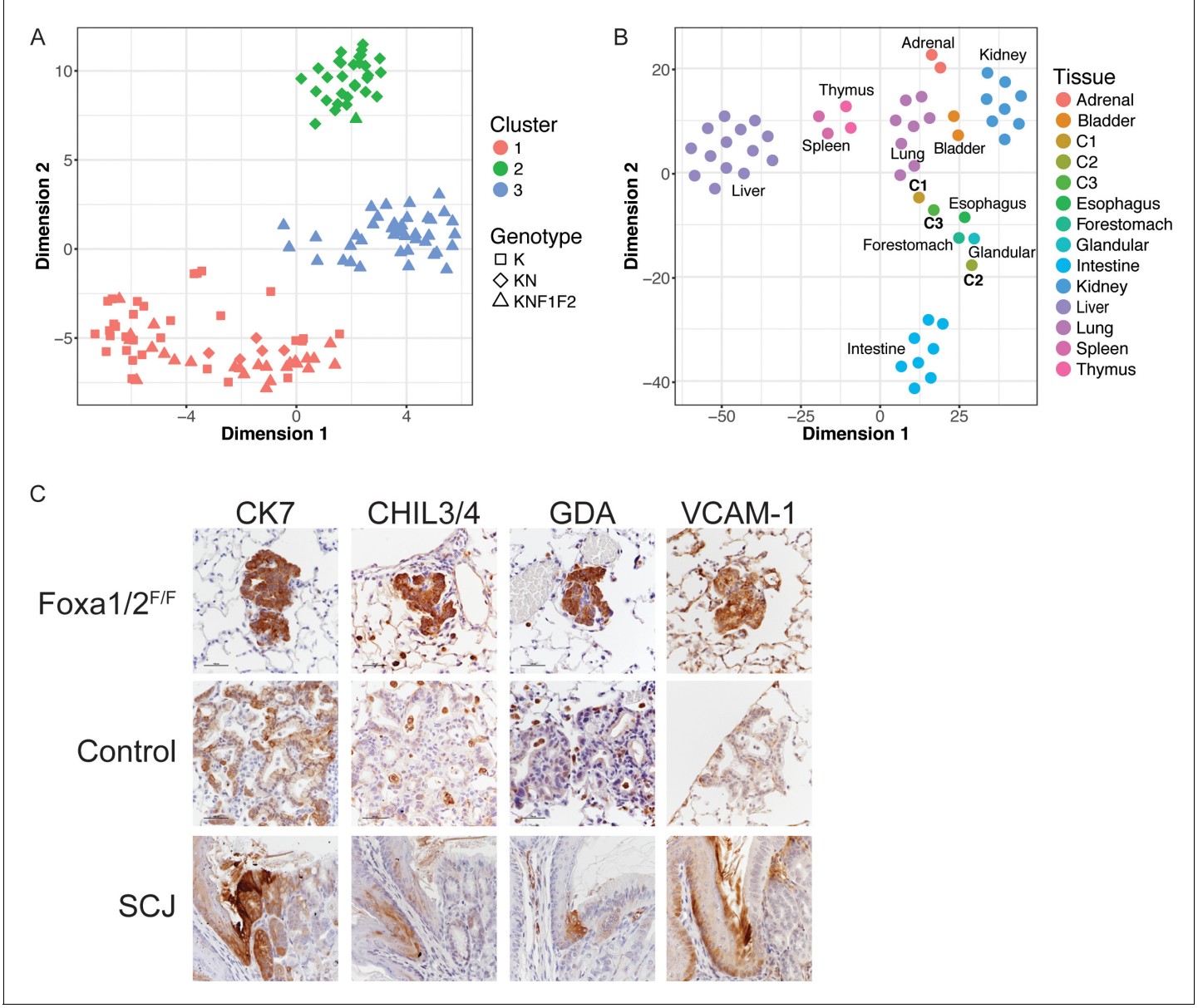

**Figure 3.** Deletion *Nkx2-1*, *Foxa1* and *Foxa2* at initiation blocks gastric differentiation and induces expression of squamocolumnar junctional markers in lung neoplasia. (**A**) tSNE plot of single-cell mRNA-Seq data derived from murine lung tumor cells (n = 134). Cells were sorted based on tdTomato expression from mice of the following genotypes: $Kras^{LSL-G12D/+}$(K), n = 1 mouse), $Kras^{LSL-G12D/+}$; $Nkx2-1^{F/F}$ (KN), n = 1 mouse), $Kras^{LSL-G12D/+}$; $Nkx2-1^{F/F}$; $Foxa1^{F/F}$; $Foxa2^{F/F}$ (KNF1F2, n = 2 mice). Color indicates cancer cell cluster. Shape indicates genotype of mouse from which cell was isolated. (**B**) tSNE plot of three cancer cell clusters (C1–C3) and a panel of normal murine tissue. 'Glandular' indicates glandular stomach. (**C**) IHC for indicated proteins in lung neoplasia 5 weeks after initiation with PGK-Cre lentivirus in $Kras^{LSL-G12D/+}$; $Nkx2-1^{F/F}$; $Foxa1^{F/F}$; $Foxa2^{F/F}$ mice and NKX2-1-negative controls. SCJ: normal squamocolumnar junction (forestomach on left, glandular stomach on right). Scale bar: 100 microns.

DOI: https://doi.org/10.7554/eLife.38579.007

The following figure supplement is available for figure 3:

**Figure supplement 1.** Deletion of *Nkx2-1*, *Foxa1* and *Foxa2* at initiation blocks gastric differentiation and induces expression of squamocolumnar junctional markers in lung neoplasia.

DOI: https://doi.org/10.7554/eLife.38579.008

performed RNA-Seq on sorted bulk tumor cells from $Kras^{LSL-G12D/+}$ and $Kras^{LSL-G12D/+}$; $Nkx2-1^{F/F}$ mice (n = 3 each, *Supplementary file 4*) and found a strong correlation (Pearson correlation coefficient = 0.62) between the differentially expressed genes identified in single cell and bulk analyses (*Figure 3—figure supplement 1A*). We also compared our single-cell datasets with published data

from other groups. We found that normal murine type two pneumocytes (*Treutlein et al., 2014*), which are the presumed cell of origin for NKX2-1-positive tumor cells, clustered with presumptive NKX2-1-positive C1 cells (*Figure 3—figure supplement 1B*). We also used principal component analysis (PCA) to compare our single-cell data with a gene signature of human IMA (*Guo et al., 2017*). In this analysis, the IMA signature caused C2 cells to cluster separately from the other cells (*Figure 3—figure supplement 1C*). This shows that C2 cells are more similar to IMA than C1 and C3, as would be expected if they represent the NKX2-1-negative tumor cell population.

## NKX2-1; FoxA1/2-deficient tumor cells express markers of the squamocolumnar junctional epithelium of the GI tract

To characterize the identity of our tumor cells in a global manner, we compared our single-cell RNA-Seq data with total RNA-Seq data from a panel of mouse tissues (*Supplementary file 5*). The 50 genes in each tissue with the highest expression compared to the other tissues in the panel were identified. Expression data for this set of genes was extracted from the single cell and tissue data-sets and evaluated using two approaches: tSNE (*Figure 3B*) and hierarchical clustering on principal components (HCPC, *Figure 3—figure supplement 1D*), which combines PCA, hierarchical clustering and k-means clustering. In both approaches, we found that C1 was most similar to normal lung, and that C2 was most similar to glandular stomach. C3 cells clustered near the upper GI tract, in particular the forestomach and esophagus. However, cosine similarity analysis (*Supplementary file 6*) showed that C3 cells are not as closely related to esophagus/forestomach as C1 and C2 are to lung and glandular stomach, respectively. This bioinformatic analysis is in consonance with microscopic evaluation of complete recombinants, which lack morphological features of a multi-layered, keratinizing squamous epithelium (*Figure 1*) that is found in the normal esophagus and forestomach. Moreover, the vast majority of complete recombinants cells do not express ΔNp63, a master regulator of squamous differentiation, or the squamous marker cytokeratin 5 (CK5) (*Figure 3—figure supplement 1E*). Thus, complete ablation of NKX2-1, FoxA1 and FoxA2 causes lung tumor cells to adopt an identity that is neither pulmonary nor gastric, but also is not fully squamous. Indeed, it appears that the exact differentiation state adopted by these cells is not well represented in the panel of tissues evaluated.

Recent studies have described a small but discrete transitional zone at the squamocolumnar junction (SCJ) of the gastrointestinal (GI) tract, just proximal to the glandular stomach, which is not included as a discrete entity in our tissue panel. This transitional zone consists of a bilayered epithelium expressing high levels of cytokeratin 7 (CK7) (*Jiang et al., 2017*; *Wang et al., 2011*), including a ΔNp63/CK5-positive basal layer and a ΔNp63/CK5-negative luminal layer. Intriguingly, complete recombinants in $Kras^{LSL-G12D/+}$; $Nkx2-1^{F/F}$; $Foxa1^{F/F}$; $Foxa2^{F/F}$ mice have uniformly high levels of CK7 protein that are comparable to the SCJ (*Figure 3C*).

Manual inspection of genes specifically expressed in C3 vs. both C1 and C2 (using both SC3 and SCDE) revealed that several of these genes are expressed at high levels at the SCJ of the GI tract and/or the cervix. These genes include *Chil4* (*Nio et al., 2004*), *Gda* and *Mmp7* (*Herfs et al., 2012*), and *Vcam1* (*Figure 3C*). Other C3-specific genes are expressed at higher levels throughout the forestomach and esophagus than glandular stomach, including *Cav1*, *Cdh13*, *Hilpda*, *Fbln2* and *Rbp7* (*Uhlén et al., 2015*). Using IHC, we found that protein levels of several these genes are much higher in complete recombinants than in NKX2-1-negative lesions (*Figure 3C* and *Figure 3—figure supplement 1E*).

These data led us to evaluate FoxA1/2 levels at the SCJ of the murine GI tract (*Figure 3—figure supplement 1F*). Both FoxA1 and FoxA2 are expressed in the glandular stomach. Interestingly, FoxA2 expression ends at the SCJ and is absent throughout the squamous forestomach and esophagus. In contrast, FoxA1 levels are very low but detectable at the SCJ then increase in the proximal forestomach and esophagus. Thus, overall FoxA1/2 levels appear to reach their nadir at the SCJ and distal forestomach of the normal murine GI tract. Taken together, our data show that FoxA1/2 are required for NKX2-1-deficient lung tumor cells to adopt a gastric identity. Moreover, concomitant loss of NKX2-1 and FoxA1/2 activity at tumor initiation leads to a distinct differentiation state characterized by expression of multiple markers of the transitional epithelium normally found at the SCJ of the GI tract.

## FoxA1/2 are downregulated in the squamous component of murine and human adenosquamous carcinoma of the lung

Although $Kras^{LSL-G12D/+}$; $Nkx2-1^{F/F}$; $Foxa1^{F/F}$ and $Kras^{LSL-G12D/+}$; $Nkx2-1^{F/F}$; $Foxa2^{F/F}$ mice exhibited minimal obvious phenotypes at early timepoints (*Figures 1–2*), we found that a subset of these mice developed macroscopic adenosquamous carcinomas (AdSCCs) at 20 weeks post-initiation (*Figure 4* and *Figure 4—figure supplement 1A*). In contrast, we did not find AdSCCs in any of the $Kras^{LSL-G12D/+}$; $Nkx2-1^{F/F}$ mice aged to 20 weeks post-initiation. Human AdSCC is an uncommon but aggressive lung cancer subtype that contains a mix of clonally related adenocarcinoma and squamous cell components (*Shu et al., 2013*; *Tochigi et al., 2011*). In our mice, AdSCCs consisted of a mucinous adenocarcinoma component that was continuous with, and typically circumscribed, a well-differentiated, keratinizing squamous cell carcinoma component (*Figure 4A–C*). Both components were tdTomato-positive, indicating that these tumors had arisen through Cre-mediated recombination (*Figure 4B*). Although both components were NKX2-1-negative, markers of gastric differentiation were restricted to the adenocarcinoma component, and markers of squamous differentiation (including ΔNp63 and cytokeratins 5 and 14 (CK5 and CK14), but not SOX2) were selectively expressed in the SCC component (*Figure 4B* and *Figure 4—figure supplement 1C*).

Given that genetic deletion of *Foxa1* and *Foxa2* at initiation completely suppressed mucinous gastric differentiation, we evaluated expression of both transcription factors in AdSCCs. In $Kras^{LSL-G12D/+}$; $Nkx2-1^{F/F}$; $Foxa2^{F/F}$ mice, we found that FoxA2 was absent in both components, whereas FoxA1 was expressed in the adenocarcinoma components but absent in the SCC (*Figure 4B*). AdSCC in the $Kras^{LSL-G12D/+}$; $Nkx2-1^{F/F}$; $Foxa1^{F/F}$ mouse exhibited the opposite pattern, that is, FoxA1 loss in both components and FoxA2 expression only in the adenocarcinoma component (*Figure 4—figure supplement 1B*). Thus, the SCC component is always associated with stochastic loss of FoxA1/2 expression. Given that $Kras^{LSL-G12D/+}$; $Nkx2-1^{F/F}$; $Foxa1^{F/F}$; $Foxa2^{F/F}$ mice contain incomplete recombinants that retain either FoxA1 or FoxA2, we also observed AdSCCs in a subset of these mice at 20 weeks (*Figure 4—figure supplement 1A*). As expected, AdSCCs in $Kras^{LSL-G12D/+}$; $Nkx2-1^{F/F}$; $Foxa1^{F/F}$; $Foxa2^{F/F}$ mice always expressed either FoxA1 or FoxA2 in the adenocarcinoma component and stochastic loss of the other paralogue in the squamous component.

These data suggest that when only one FoxA paralogue is expressed in mucinous lung adenocarcinoma, stochastic loss of the other FoxA paralogue can occur as the tumors progress. This stochastic loss of FoxA activity is associated with a profound change in differentiation state, with FoxA1/2-negative cells upregulating a keratinizing squamous differentiation program. This is in sharp contrast to the differentiation state of tumor cells in which FoxA1/2 loss was engineered at the time of tumor initiation (*Figure 3*), which led to an SCJ-like phenotype. These results raise the possibility that the genetic and/or epigenetic context in which FoxA1/2 activity is lost may have a significant influence on the cellular identity adopted by lung tumor cells.

We next analyzed FoxA1/2 expression by IHC in human AdSCC (n = 12) to determine whether these transcription factors are differentially expressed between adenocarcinoma and squamous components (*Figure 4D–E*). FoxA1 and FoxA2 were expressed in the adenocarcinoma component of all cases. In half of the cases, FoxA1 and FoxA2 were both downregulated in the squamous component (n = 5 cases with complete loss of expression and n = 1 case with detectable but diminished expression). In the other half, either FoxA1 (n = 5) or FoxA2 (n = 1) exhibited downregulation in the squamous component. Thus, half of the human AdSCC examined exhibit the same pattern of FoxA1/2 downregulation that we observe in our mouse model. Moreover, all cases exhibit at least partial reduction in expression of FoxA1 or FoxA2 in the squamous component. Taken together, these data suggest that reduced FoxA activity is commonly associated with adenosquamous transdifferentiation in human lung cancer.

## Context-dependent induction of squamous cell carcinoma by loss of FoxA1/2

To test the hypothesis that loss of FoxA1/2 activity might promote squamous differentiation only in specific contexts, we generated $Kras^{FSF-G12D/+}$; $Rosa^{FSF-CreERT2}$; $Nkx2-1^{F/F}$; $Foxa1^{F/F}$; $Foxa2^{F/F}$ mice as well as controls wild type for either *Foxa1* or both *Foxa1* and *Foxa2*. In these mice, delivery of the FlpO recombinase (via Ad5CMV-FlpO adenovirus) to the lung epithelium activates transcription of the $Kras^{G12D}$ oncogene from its endogenous locus (*Young et al., 2011*) and transcription of Cre[ERT2]

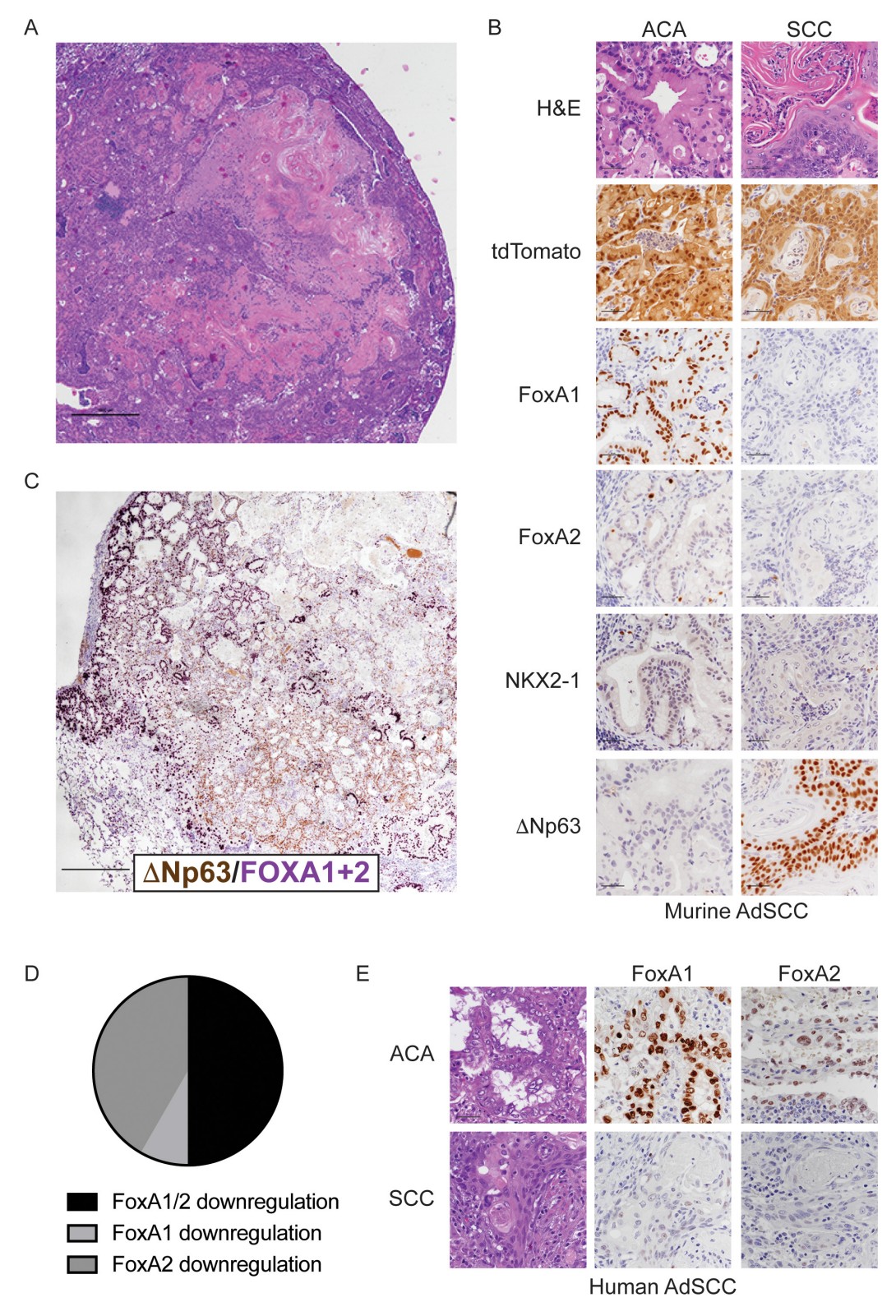

**Figure 4.** FoxA1 and FoxA2 are downregulated in the squamous component of murine and human adenosquamous lung carcinoma. (A) H and E of AdSCC arising in $Kras^{LSL-G12D/+}$; $Nkx2-1^{F/F}$; $Foxa2^{F/F}$ mouse. Scale bar: 1000 microns. (B) H and E and IHC of adenocarcinoma (left) and squamous (right) components of AdSCC arising in $Kras^{LSL-G12D/+}$; $Nkx2-1^{F/F}$; $Foxa2^{F/F}$ mouse. Scale bar: 100 microns. (C) Dual IHC for ΔNp63 (brown) and FoxA1/2 (purple) in AdSCC arising in $Kras^{LSL-G12D/+}$; $Nkx2-1^{F/F}$; $Foxa2^{F/F}$ mouse. Scale bar: 1000 microns. (D) Percent of human AdSCC cases (n = 12) exhibiting
*Figure 4 continued on next page*

*Figure 4 continued*

downregulation of FoxA1 and/or FoxA2 expression in the SCC component as assessed by IHC. (E) Representative IHC for FoxA1 and FoxA2 in a human AdSCC exhibiting downregulation of both proteins in the SCC component. Scale bar: 100 microns.

DOI: https://doi.org/10.7554/eLife.38579.009

The following figure supplement is available for figure 4:

**Figure supplement 1.** FoxA1 and FoxA2 are downregulated in the squamous component of murine and human adenosquamous lung carcinoma.

DOI: https://doi.org/10.7554/eLife.38579.010

from the *Rosa26* locus (*Schönhuber et al., 2014*). Tamoxifen is then used to activate the Cre[ERT2] protein and drive recombination of lineage specifiers in KRAS[G12D]-expressing lung neoplasia. To determine whether loss of NKX2-1, FoxA1 and FoxA2 in established neoplasia was sufficient to induce full squamous differentiation, we administered tamoxifen 1 week after tumor initiation with Ad5CMV-FlpO, then analyzed tumors 4 weeks later (outline in *Figure 5A*).

Histopathologic analysis of controls showed that the lungs contained mucinous adenocarcinoma that expressed HNF4α and the expected pattern of FoxA1/2 (*Figure 5B* and *Figure 5—figure supplement 1A*). Almost all lesions in *Nkx2-1^F/F* and *Nkx2-1^F/F; Foxa2^F/F* mice were ΔNp63-negative (*Figure 5B* and *Figure 5—figure supplement 1C*). Indeed, only one mouse in each control group exhibited a single lesion with ΔNp63-positive cells. In contrast, all *Kras^FSF-G12D/+; Rosa^FSF-CreERT2; Nkx2-1^F/F; Foxa1^F/F; Foxa2^F/F* mice harbored numerous non-mucinous lesions lacking FoxA1/2 and HNF4α. Most of these lesions were morphologically similar to the SCJ-like lesions generated with Cre-mediated recombination at tumor initiation (*Figure 1*).

Strikingly, four out of eight *Kras^FSF-G12D/+; Rosa^FSF-CreERT2; Nkx2-1^F/F; Foxa1^F/F; Foxa2^F/F* mice harbored well-differentiated squamous cell carcinomas (SCCs) characterized by a stratified squamous epithelium with extensive keratinization (*Figure 5B* and *Figure 5—figure supplement 1B*). In contrast to the AdSCCs that arise stochastically from mucinous adenocarcinomas (*Figure 4*), these SCCs appeared to be discrete lesions and were not surrounded by HNF4α-positive mucinous adenocarcinoma (*Figure 5B*). As expected, all SCCs in this model expressed ΔNp63 (*Figure 5B*). Interestingly, we even detected ΔNp63 in a significant minority of non-keratinizing lesions in these mice (*Figure 5B*), which contrasts with the lack of ΔNp63 expression in the complete recombinants of *Kras^LSL-G12D/+; Nkx2-1^F/F; Foxa1^F/F; Foxa2^F/F* mice (*Figure 3—figure supplement 1E*).

Most of the microscopic analysis of *Kras^LSL-G12D/+; Nkx2-1^F/F; Foxa1^F/F; Foxa2^F/F* mice (*Figures 1–4*) was performed on lesions generated with lentivirus expressing Cre under the control of the *Pgk* promoter. To control for the possibility that the use of adenovirus and/or the CMV promoter might have played a role in the phenotypes observed with sequential recombination, we infected *Kras^LSL-G12D/+; Nkx2-1^F/F; Foxa1^F/F; Foxa2^F/F* mice (n = 6) with Ad5CMV-Cre and harvested tumors 5 weeks after infection. Importantly, none of the mice harbored SCCs, despite the presence of multiple complete recombinants (*Figure 5—figure supplement 1C*). Interestingly, ΔNp63 expression was slightly higher in these lesions than in lesions from mice of the same genotype infected with Pgk-Cre lentivirus (data not shown).

Taken together, these data identify a specific context in which loss of FoxA1/2 activity is sufficient to induce full squamous differentiation in the lung. Since FoxA1/2 loss was induced only 1 week after KRAS[G12D] expression in this experiment, it seems likely that enhanced competence for squamous differentiation is a direct result of KRAS[G12D] expression rather than stochastic genetic alterations accruing over time. Moreover, the fact that only a subset of neoplastic lesions are keratinizing SCC raises the possibility that a specific subpopulation of lung epithelial cells may exhibit enhanced competence for squamous differentiation in this system.

## Squamous cell carcinoma arises from SPC-positive lung epithelial cells

To define more precisely the cell type from which SCCs arise in the sequential recombination model, we generated an adenovirus in which expression of the FlpO recombinase is driven by the murine SPC promoter. This promoter has been extensively validated to drive Cre expression primarily in type 2 pneumocytes of the alveoli (*Sutherland et al., 2011*). To validate this promoter in our sequential recombination system, we generated *Kras^FSF-G12D/+; Rosa^FSF-CreERT2* harboring a *CAG-LSL-HA-UPRT* transgene (*Gay et al., 2013*), in which the HA-tagged UPRT enzyme is only expressed

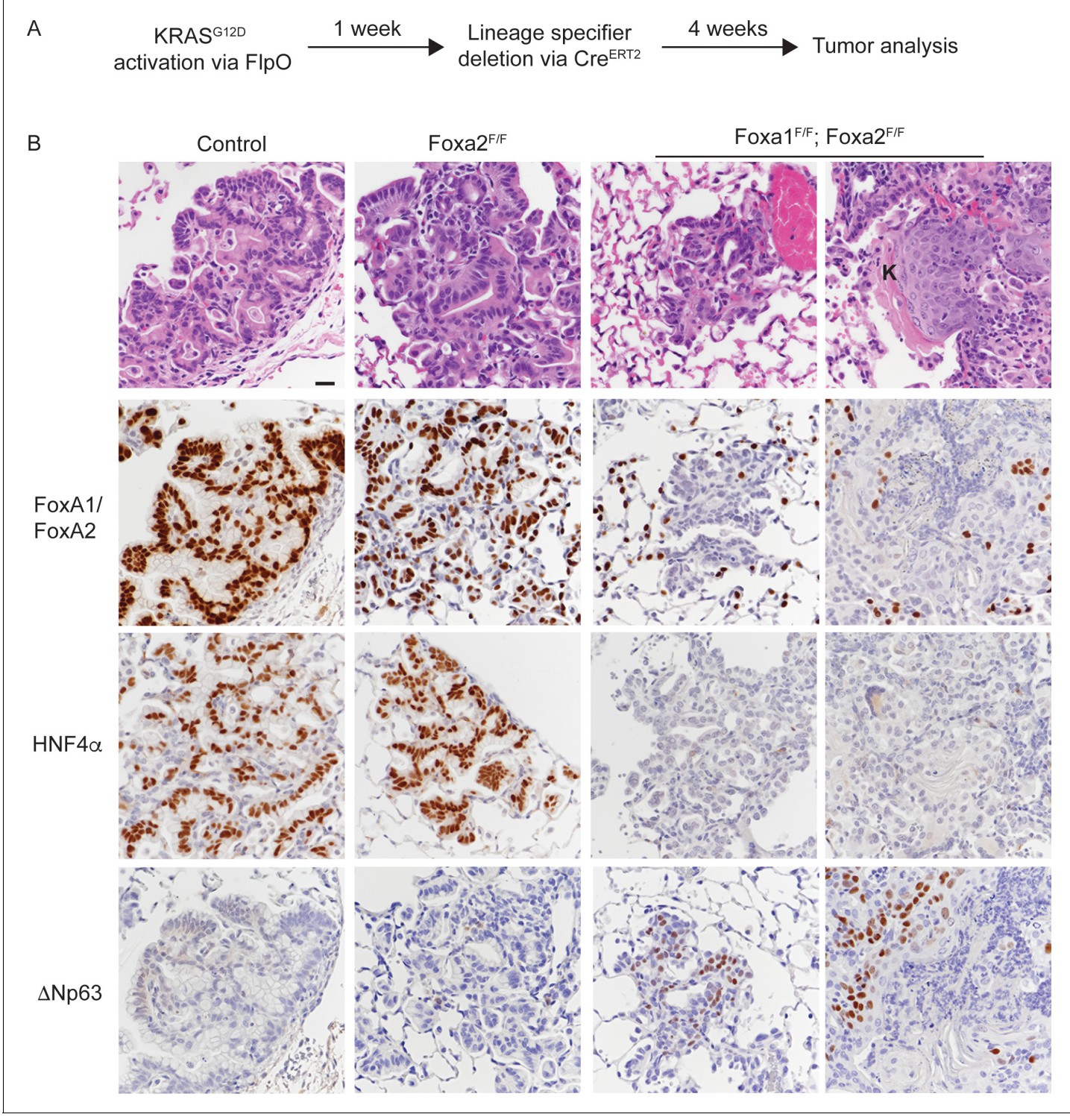

**Figure 5.** Uncoupling KRAS[G12D] activation from lineage specifier deletion promotes squamous cell carcinoma formation in the lung. (**A**) Schematic of experimental design. (**B**) H and E and IHC for indicated proteins in tumors from mice harboring the conditional alleles *Kras[FSF-G12D/+]; Rosa[FSF-CreERT2]; Nkx2-1[F/F]* (controls) alone and in combination with either *Foxa2[F/F]* or *Foxa1[F/F]; Foxa2[F/F]*. 'K' indicates acellular keratin. All mice were given tamoxifen 1 week after tumor initiation. Tamoxifen administration consisted of six intraperitoneal doses over nine days, followed by tamoxifen-containing chow until the end of the experiment. Scale bar: 50 microns.

DOI: https://doi.org/10.7554/eLife.38579.011

The following figure supplement is available for figure 5:

*Figure 5 continued on next page*

Figure 5 continued

**Figure supplement 1.** Uncoupling KRAS^G12D activation from lineage specifier deletion promotes squamous cell carcinoma formation in the lung.
DOI: https://doi.org/10.7554/eLife.38579.012

after Cre-based recombination of the STOP cassette. These mice were infected with Ad5-SPC-FlpO or Ad5-CMV-FlpO, treated with tamoxifen 1-week post-infection, and subjected to histopathologic analysis 3 weeks post-infection. Whereas recombination was readily detectable in the bronchioles and alveoli of Ad5-CMV-FlpO infected mice, recombination was restricted to the alveoli of Ad5-SPC-FlpO infected mice (*Figure 6—figure supplement 1A*).

Next, we infected a cohort of $Kras^{FSF-G12D/+}$; $Rosa^{FSF-CreERT2}$; $Nkx2-1^{F/F}$; $Foxa1^{F/F}$; $Foxa2^{F/F}$ mice, along with $Kras^{FSF-G12D/+}$; $Rosa^{FSF-CreERT2}$; $Nkx2-1^{F/F}$ controls with Ad5-SPC-FlpO. As an additional control, we infected a group of $Kras^{LSL-G12D/+}$; $Rosa^{LSL-tdTomato}$; $Nkx2-1^{F/F}$; $Foxa1^{F/F}$; $Foxa2^{F/F}$ mice with Ad5-SPC-Cre. All mice were treated with tamoxifen 1 week after infection and analyzed 5 weeks after infection. As expected, the lungs of $Kras^{FSF-G12D/+}$; $Rosa^{FSF-CreERT2}$; $Nkx2-1^{F/F}$ controls harbored numerous mucinous lesions in the alveoli that expressed FoxA1/2 and lacked squamous markers such as ΔNp63 and CK5 (*Figure 6*, left panel). $Kras^{FSF-G12D/+}$; $Rosa^{FSF-CreERT2}$; $Nkx2-1^{F/F}$; $Foxa1^{F/F}$; $Foxa2^{F/F}$ mice harbored FoxA1/2-negative lesions of two distinct morphologies (*Figure 6*, central panels). All mice harbored SCJ-like lesions that were predominantly CK7-positive/CK5-negative and expressed ΔNp63 in a minority of cells. Moreover, 63% of these mice (n = 5 out of 8) harbored well-differentiated, keratinizing SCCs that were CK7-negative/CK5-positive and expressed ΔNp63 (*Figure 6*, central panels and *Figure 6—figure supplement 1B*). Overall, these phenotypes were very similar to those observed when tumors were initiated with Ad5-CMV-FlpO in these mice (*Figure 5*). Importantly, we did not identify SCC in any of the $Kras^{LSL-G12D/+}$; $Rosa^{LSL\_tdTomato}$; $Nkx2-1^{F/F}$; $Foxa1^{F/F}$; $Foxa2^{F/F}$ mice infected with Ad5-SPC-Cre. These mice harbored CK7-positive/CK5-negative SCJ-like lesions that were essentially identical to lesions initiated by lentivirus in previous experiments (*Figures 1–3*).

Taken together, these data show that loss of NKX2-1, FoxA1, and FoxA2 in SPC-positive alveolar cells has distinct outcomes depending on the state of oncogenic signaling in these cells. When these lineage specifiers are lost in normal SPC-positive cells (concomitant with KRAS^G12D activation), the resulting neoplasia equilibrates to a uniform SCJ-like state marked by a CK7-positive/CK5-negative immunophenotype. In contrast, SPC-positive cells that have already been subjected to oncogenic signaling from KRAS^G12D for ~1 week have the potential to undergo full squamous transdifferentiation (CK7-negative/CK5-positive) and become well-differentiated keratinizing SCCs.

## Discussion

Lung adenocarcinomas can adopt a variety of differentiation states, and changes in cellular identity can have both prognostic and therapeutic implications for patients with this disease. We have previously shown that engineered loss of the pulmonary lineage specifier NKX2-1 causes lung adenocarcinoma cells to shed their pulmonary identity and adopt a gastric differentiation state that is also observed in human IMA (*Snyder et al., 2013*). Here, we show that FoxA1 and FoxA2 are required for lung adenocarcinomas to adopt a mucinous, gastric differentiation state in the absence of NKX2-1. Although FoxA1/2 can regulate lung adenocarcinoma biology individually in some contexts (*Li et al., 2015*), their functional redundancy in IMA is consistent with their frequently redundant role in endodermal tissue specification (reviewed in *Golson and Kaestner, 2016*). The precise mechanisms by which FoxA1/2 specifically activate a gastric program in NKX2-1 negative lung cancer, as opposed to other potential endodermal differentiation states (e.g. hepatic, pancreatic, lower GI tract etc.), remain to be determined. However, it appears likely that FoxA1/2 regulate gastrointestinal differentiation programs in other types of cancer. For example, pancreatic ductal adenocarcinoma (PDAC) and its precursors often express many of the same foregut markers as NKX2-1-negative lung adenocarcinoma (*Tata et al., 2018*; *Bailey et al., 2016*; *Prasad et al., 2005*). FoxA1/2 levels are much higher in the subset of PDAC expressing a foregut differentiation program than in those tumors with a more mesenchymal/squamous differentiation state (*Bailey et al., 2016*). In addition, aberrant activation of a gastrointestinal differentiation program in prostate cancer, which can mediate castration resistance, is driven by HNF4γ in cooperation with FoxA1 (*Shukla et al., 2017*).

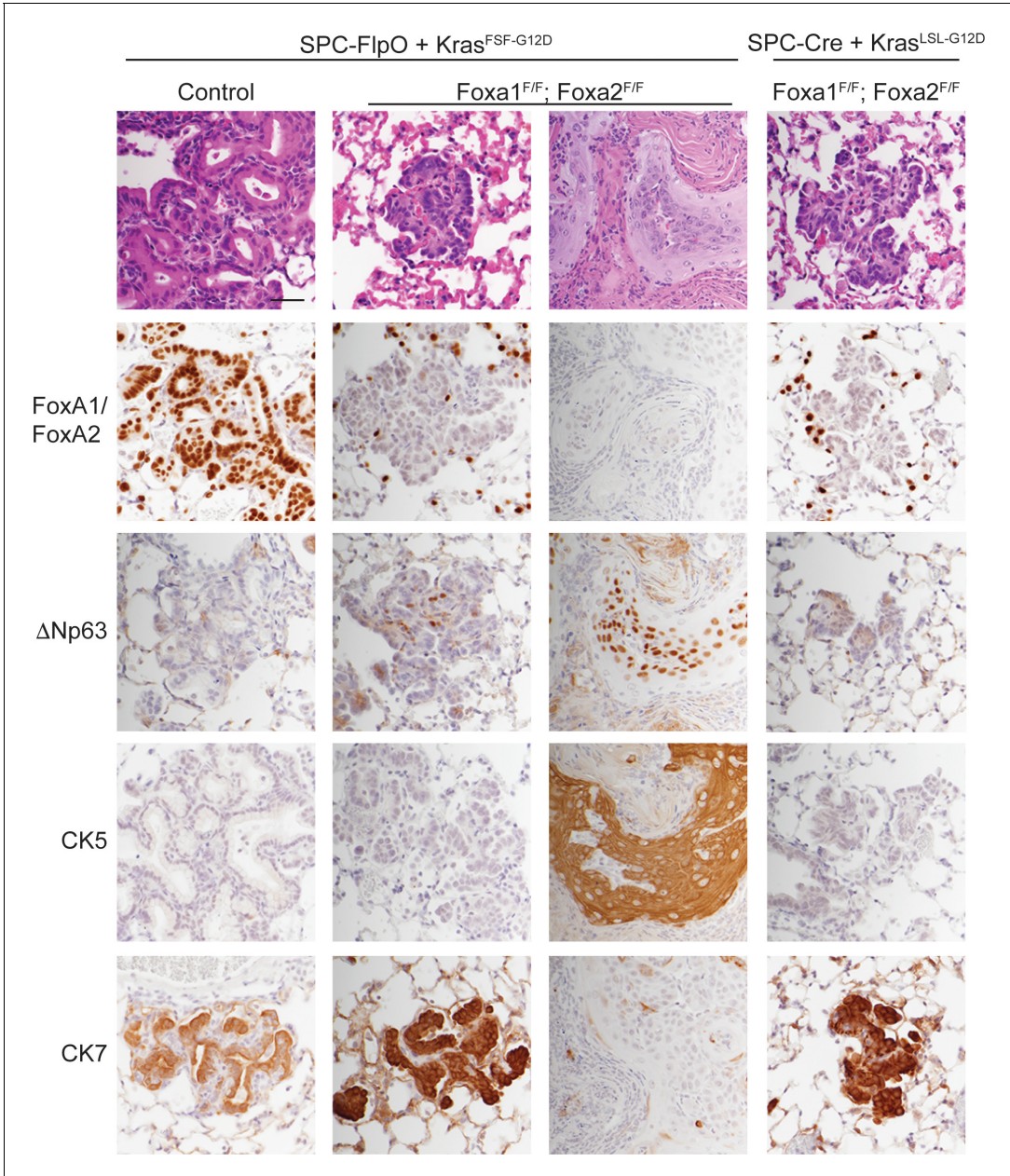

**Figure 6.** SPC-positive cells give rise to squamous cell carcinoma when KRAS[G12D] activation is uncoupled from lineage specifier deletion. H and E and IHC for indicated proteins in neoplasia 5 weeks post initiation. In mice harboring the conditional alleles $Kras^{FSF-G12D/+}$; $Rosa^{FSF-CreERT2}$; $Nkx2-1^{F/F}$ (controls) alone and in combination with $Foxa1^{F/F}$; $Foxa2^{F/F}$, lung tumors were in initiated with Ad5-SPC-FlpO adenovirus. In mice harboring the conditional alleles $Kras^{LSL-G12D/+}$; $Rosa^{LSL-tdTomato}$; $Nkx2-1^{F/F}$; $Foxa1^{F/F}$; $Foxa2^{F/F}$, tumors were initiated with Ad5-SPC-Cre (right column). All mice were given tamoxifen 1 week after tumor initiation. Tamoxifen administration consisted of four intraperitoneal doses over 5 days, followed by tamoxifen-containing chow until the end of the experiment. Scale bar: 100 microns.

DOI: https://doi.org/10.7554/eLife.38579.013

The following figure supplement is available for figure 6:

**Figure supplement 1.** SPC-positive cells give rise to squamous cell carcinoma when KRAS[G12D] activation is uncoupled from lineage specifier deletion.
DOI: https://doi.org/10.7554/eLife.38579.014

Interestingly, the precise consequences of FoxA1/2 loss in lung cancer are highly dependent on the specific context in which it occurs (model, *Figure 7*). When *Nkx2-1*, *Foxa1* and *Foxa2* are deleted at tumor initiation, the resulting lung lesions lacked evidence of either pulmonary or gastric differentiation (*Figure 3*). Instead, complete recombinants expressed several genes enriched at the SCJ of

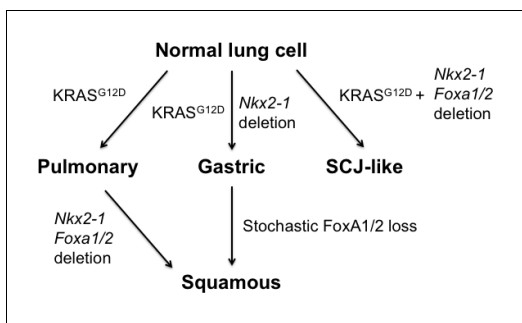

**Figure 7.** Model of context-specific regulation of lung cancer identity by NKX2-1, FoxA1 and FoxA2. SCJ: squamocolumnar junction of GI tract.

DOI: https://doi.org/10.7554/eLife.38579.015

the GI tract, which contains a small but distinct non-keratinized transitional columnar epithelium marked by high levels of CK7 (*Jiang et al., 2017*). The transitional epithelium consists of a ΔNp63-positive basal progenitor layer, which can give rise to Barrett's metaplasia, and a differentiated ΔNp63-negative luminal layer. Thus, in the absence of FoxA1/2 activity, *Nkx2-1* deletion causes normal lung epithelial cells to adopt a cell fate resembling the transitional epithelium that localizes immediately proximal to the glandular stomach. Given the lack of ΔNp63 expression in complete recombinants, we speculate that these tumor cells more closely resemble the luminal cells of the transitional epithelium, which may account in part for their limited proliferative capacity.

In contrast, both stochastic (*Figure 4*) and engineered (*Figures 5–6*) loss of FoxA1/2 in established KRAS$^{G12D}$-driven lesions initiated in SPC-positive cells was accompanied by activation of a robust squamous differentiation program, as evidenced by a stratified multi-layered epithelium with extensive keratinization and expression of ΔNp63 and CK5. These data suggest that signaling from KRAS$^{G12D}$ enhances the capacity of SPC-positive cells to activate a squamous differentiation program in the absence of NKX2-1 and FoxA1/2. Additional studies will be needed to determine the mechanism(s) that account for this enhanced propensity for squamous differentiation. ΔNp63 is an activator of squamous differentiation and is generally thought to function as an oncogene in SCC (*Watanabe et al., 2014*), so the increased levels of ΔNp63 when *Nkx2-1;Foxa1/2* deletion occurs in established lesions (*Figures 5–6*) vs. at tumor initiation (*Figure 3*) are likely to be one major factor that dictates whether cells adopt an SCJ-like vs. SCC fate. We speculate that signaling from KRAS$^{G12D}$ either alters the epigenetic state of elements regulating ΔNp63 directly, or influences the activity of its numerous upstream regulators (*Yoh and Prywes, 2015*). It is also unclear why SPC-positive cells adopt two distinct fates in our sequential mutagenesis experiments (SCJ-like vs SCC, *Figure 6*). Intrinsic heterogeneity of the SPC-positive population (prior to KRAS$^{G12D}$ expression) could account for this observation (*Kim et al., 2005*; *Nabhan et al., 2018*; *Zacharias et al., 2018*). Alternatively, heterogeneous response to KRAS$^{G12D}$ signaling could also play a role. Proliferation rate can influence changes in cell identity (*Soufi and Dalton, 2016*), and it is possible that only a subset of SPC-positive cells are actively cycling one week after KRAS$^{G12D}$ expression. Regardless, the fact that SPC-positive cells readily give rise to SCC contrasts with other investigations of cell type specificity in mouse models of SCC. In *Kras$^{G12D}$; Lkb1* conditional mice CC10-positive lung epithelial cells are the predominant cell of origin for adenosquamous carcinomas, whereas SPC-positive cells mainly give rise to adenocarcinomas (*Nagaraj et al., 2017*; *Zhang et al., 2017*). In other murine models driven by SOX2 expression and either deletion of *Pten* and *Cdkn2ab* (*Ferone et al., 2016*) or *Nkx2-1* (7), both cell types can give rise to SCCs, although CC10-positive cells appeared to do so more efficiently.

Context also appears to be critical for the effect of FoxA1/2 loss on tumor growth. We have previously shown that *Nkx2-1* deletion augments KRAS$^{G12D}$-driven tumorigenesis (*Snyder et al., 2013*). Concomitant *Foxa1/2* deletion at initiation reverses this phenotype (*Figure 2*), showing that when FoxA1/2 are absent at tumor initiation, NKX2-1-negative lesions equilibrate to a low proliferation state that never progresses to macroscopic disease. However, the stochastic emergence of macroscopic FoxA1/2-negative AdSCCs (*Figure 4*) argues that there is a selective advantage to loss of FoxA1/2 in some established lung neoplasia. This is further reinforced by the observation that a subset of human AdSCCs downregulate FoxA1/2 in their squamous component (*Figure 4*). An apparently dichotomous and context-specific contribution of FoxA1/2 to malignant potential has been observed in tumors from other tissues (reviewed in *Golson and Kaestner, 2016*). For example, one study of human lung SCC reported that 43% of cases lacked FoxA1 expression by IHC, and that FoxA1 positivity was significantly correlated with unfavorable survival (*Deutsch et al., 2012*). In PDAC, FoxA1 can promote metastasis (*Roe et al., 2017*), despite the fact low levels of FoxA1/2 (as

well as other lineage specifiers associated with endodermal differentiation) are found in the subtype of pancreatic ductal adenocarcinoma that confers the worst prognosis (*Bailey et al., 2016*). A comprehensive evaluation of FoxA1/2 loss at distinct stages of tumorigenesis will be needed to delineate fully its context-specific role in lung tumor growth.

In summary, this work expands our understanding of the lineage specifiers that coordinately regulate the growth and identity of lung adenocarcinoma. We show that FoxA1 and FoxA2 regulate the growth and gastric identity of NKX2-1-negative lung adenocarcinoma. In the absence of FoxA1/2 activity, NKX2-1-negative tumor cells adopt a more proximal cell fate with features of either the transitional epithelium of the SCJ or the squamous epithelium of the forestomach/esophagus, depending on the context of FoxA1/2 loss. Squamous transdifferentiation has been linked to drug resistance in human lung adenocarcinomas (*Hou et al., 2017*), and it will be interesting to determine whether FoxA1/2 downregulation plays a role in this process. More broadly, our results show that the effects of lineage specifier inactivation in cancer can be highly context-dependent, and provide an experimental system for future work to elucidate the mechanistic basis for this specificity.

# Materials and methods

**Key resources table**

| Reagent type (species) or resource | Designation | Source or reference | Identifiers | Additional information |
|---|---|---|---|---|
| Genetic reagent (*Mus musculus*) | $Kras^{LSL-G12D}$ | PMID: 11751630 | | Dr. Tyler Jacks (Massachusetts Institute of Technology, Cambridge, Massachusetts) |
| Genetic reagent (*M. musculus*) | $Kras^{FSF-G12D}$ | PMID: 21512139 | RRID:MGI: 5007794 | Dr. Tyler Jacks (Massachusetts Institute of Technology , Cambridge, Massachusetts) |
| Genetic reagent (*M. musculus*) | $Rosa26_{LSL-tdTomato}$ | PMID: 20023653 | RRID:MGI: 4436847 | Jackson Laboratories (Bar Harbor, Maine) |
| Genetic reagent (*M. musculus*) | $Rosa26_{FSF-CreERT2}$ | PMID: 25326799 | | Dr. Dieter Saur (Technische Universität München, München, Germany) |
| Genetic reagent (*M. musculus*) | $Nkx2-1^{F/F}$ | PMID: 16601074 | RRID:MGI: 3653706 | Dr. Shioko Kimura (National Cancer Institute (NCI), National Institutes of Health, Bethesda, Maryland) |
| Genetic reagent (*M. musculus*) | $Foxa1^{F/F}$ | PMID: 19141476 | RRID:MGI: 3831163 | Dr. Klaus H. Kaestner (University of Pennsylvania School of Medicine, Philadelphia, Pennsylvania, USA) |
| Genetic reagent (*M. musculus*) | $Foxa2^{F/F}$ | PMID: 10866673 | RRID:MGI: 2177357 | Dr. Klaus H. Kaestner (University of Pennsylvania School of Medicine, Philadelphia, Pennsylvania, USA) |

*Continued on next page*

*Continued*

| Reagent type (species) or resource | Designation | Source or reference | Identifiers | Additional information |
|---|---|---|---|---|
| Genetic reagent (*M. musculus*) | *CAG-LSL-HA-UPRT* | PMID: 23307870 | | Jackson Laboratories (Bar Harbor, Maine) |
| Cell line | 293T | PMID: 19561589 | | |
| Antibody | Rat monoclonal anti-BrdU | Abcam | Cat. #: ab6326, RRID: AB_305426 | IHC (1:100) |
| Antibody | Rabbit monoclonal anti-Cadherin 13 | Abcam | Cat. #: ab167407 | IHC (1:250) |
| Antibody | Rabbit polyclonal anti-Cathepsin E | Lifespan Biosciences | Cat. #: LS-B523, RRID: AB_2087236 | IHC (1:12000) |
| Antibody | Rabbit monoclonal anti-Caveolin 1 | Abcam | Cat. #: ab192869 | IHC (1:4000) |
| Antibody | Rabbit monoclonal anti-CHIL3/4 | Abcam | Cat. #: ab192029 | IHC (1:20000) |
| Antibody | Rabbit monoclonal anti-Cleaved-caspase 3 | Cell Signaling Technology | Cat. #: 9664 | IHC (1:800) |
| Antibody | Rabbit monoclonal anti-Cytokeratin-5 | Abcam | Cat #: ab52635 (EP1691Y) | IHC (1:400) |
| Antibody | Rabbit monoclonal anti-Cytokeratin-7 | Abcam | Cat #: ab181598 (EP17078) | IHC (1:20,000) |
| Antibody | Rat monoclonal anti-Cytokeratin-8 | Developmental Studies Hybridoma Bank | Cat. #: TROMA-I, RRID: AB_531826 | IHC (1:100) |
| Antibody | Rabbit monoclonal anti-Cytokeratin-14 | Abcam | Cat. #: ab181595 (EPR17350) | IHC (1:4000) |
| Antibody | Rabbit monoclonal anti-FoxA1 | Abcam | Cat. #: ab173287 | IHC (1:4000) |
| Antibody | Rabbit monoclonal anti-FoxA2 | Abcam | Cat. #: ab108422, RRID:AB_11157157 | IHC (1:1200) |
| Antibody | Goat polyclonal anti-Galectin 4 | R and D Systems | Cat. #: AF2128, RRID:AB_2297050 | IHC (1:400) |
| Antibody | Mouse monoclonal anti-Gastrokine 1 | Abnova | Cat. #: H00056287-M01, RRID:AB_1505437 | IHC (1:50) |
| Antibody | Rabbit monoclonal anti-GDA | Abcam | Cat. #: ab210606 | IHC (1:5000) |
| Antibody | Rabbit monoclonal anti-HNF4α | Cell Signaling Technology | Cat. #: 3113S, RRID:AB_2295208 | IHC (1:500) |
| Antibody | Rabbit monoclonal anti-Ki67 | Abcam | Cat. #: ab16667, RRID:AB_302459 | IHC (1:100) |

*Continued on next page*

*Continued*

| Reagent type (species) or resource | Designation | Source or reference | Identifiers | Additional information |
|---|---|---|---|---|
| Antibody | Rabbit polyclonal anti-MCM2 | Abcam | Cat. #: ab31159, RRID:AB_881276 | IHC (1:800) |
| Antibody | Polyclonal goat anti-MMP7 | R and D Systems | Cat. #: AF2967, RRID: AB_664120 | IHC (1:400) |
| Antibody | Mouse monoclonal anti-Muc5AC | Abnova | Cat. #: MAB13117 | IHC (1:100) |
| Antibody | Rabbit monoclonal anti-NKX2-1 | Abcam | Cat. #: ab76013, RRID: AB_1310784 | IHC (1:2000) |
| Antibody | Mouse monoclonal anti-p40 (ΔNp63) | Biocare Medical | Cat. #: ACI 3066 C | IHC (1:100) |
| Antibody | Mouse monoclonal anti-PDX1 | Developmental Studies Hybridoma Bank | Cat. #: F109-D12, RRID:AB_1157903 | IHC (1:10) |
| Antibody | Rat monoclonal anti-PIGR | Abcam | Cat. #: ab170321 | IHC (1:400) |
| Antibody | Rabbit monoclonal anti-PK-LR | Abcam | Cat. #: ab171744 | IHC (1:500) |
| Antibody | Rabbit polyclonal anti-proSPC | Millipore | Cat. #: AB3786, RRID: AB_91588 | IHC (1:4000) |
| Antibody | Rabbit polyclonal anti-RFP | Rockland | Cat. #: 600-401-379 | IHC (1:400) |
| Antibody | Rabbit monoclonal anti-SOX2 | Cell Signaling Technology | Cat. #: 3728, RRID: AB_2194037 | IHC (1:250) |
| Antibody | Rabbit monoclonal anti-VCAM1 | Abcam | Cat. #: ab134047, RRID:AB_2721053 | IHC (1:1000) |
| Recombinant DNA reagent | Ad5-CMVCre | Gene Transfer Vector Core, University of Iowa, IA | VVC-U of Iowa-5-HT | |
| Recombinant DNA reagent | Ad5-CMVFlpo | Gene Transfer Vector Core, University of Iowa, IA | VVC-U of Iowa-530HT | |
| Recombinant DNA reagent | Ad5-SPC-Cre | Gene Transfer Vector Core, University of Iowa, IA | VVC-Berns-1168 | |
| Recombinant DNA reagent | Ad5-SPC-FlpO | Gene Transfer Vector Core, University of Iowa, IA | VVC-Snyder-6695 | |
| Recombinant DNA reagent | PGK-Cre | PMID: 19561589 | | |
| Recombinant DNA reagent | VSVg | PMID: 19561589 | | |
| Recombinant DNA reagent | Δ8.9 | PMID: 19561589 | | |

*Continued on next page*

*Continued*

| Reagent type (species) or resource | Designation | Source or reference | Identifiers | Additional information |
|---|---|---|---|---|
| Recombinant DNA reagent | SPC-FlpO shuttle plasmid | this paper | | |
| Chemical compound, drug | Tamoxifen | Sigma-Aldrich | T5648-5G | |
| Chemical compound, drug | Tamoxifen supplemented chow | Envigo | TD.130858 | 500 mg/kg of diet |
| Chemical compound, drug | 1X PBS | ThermoFisher Scientific | 20012050 | |
| Chemical compound, drug | Trizol | ThermoFisher Scientific | 15596026 | |
| Commercial assay, kit | Bloxall | Vector Laboratories | SP-6000 | |
| Commercial assay, kit | Horse serum | Vector Laboratories | S-2012 | |
| commercial assay, kit | Rodent Block M | Biocare Medical | RBM961 | |
| Commercial assay, kit | ImmPRESS anti-rabbit HRP | Vector Laboratories | MP-7401 | |
| Commercial assay, kit | ImmPRESS anti-rat HRP | Vector Laboratories | MP-7444 | |
| Commercial assay, kit | ImmPRESS anti-goat HRP | Vector Laboratories | MP-7405 | |
| Commercial assay, kit | Anti-mouse secondary | Biocare Medical | MM620 | |
| Commercial assay, kit | ImmPACT DAB Peroxidase (HRP) Substrate | Vector Laboratories | SK-4105 | |
| Commercial assay, kit | ImmPACT VIP Peroxidase (HRP) Substrate | Vector Laboratories | SK-4605 | |
| Commercial assay, kit | Hematoxylin | Fisher Scientific | 6765003 | |
| Commercial assay, kit | Collagenase type I | ThermoFisher Scientific | 17100017 | |
| Commercial assay, kit | Elastase | Worthington Biochemical Corporation | LS002280 | |
| Commercial assay, kit | Dispase | Corning | 354235 | |
| Commercial assay, kit | Deoxyribonuclease I | Sigma-Aldrich | DN25 | |
| Commercial assay, kit | Red Blood Cell Lysis Buffer | eBioscience | 00-4333-57 | |
| Commercial assay, kit | PureLink RNA Mini kit | ThermoFisher Scientific | 12183018A | |

## Mice and tumor initiation

Mice harboring *Kras*$^{LSL-G12D}$ (**Jackson et al., 2001**), *Kras*$^{FSF-G12D}$ (**Young et al., 2011**), *Rosa26*$^{LSL-tdTomato}$ (**Madisen et al., 2010**), *Rosa26*$^{FSF-CreERT2}$ (30), *Nkx2-1*$^{F/F}$ (**Kusakabe et al., 2006**), *Foxa1*$^{F/F}$ (**Gao et al., 2008**), *Foxa2*$^{F/F}$ (**Sund et al., 2000**) and CAG-LSL-HA-UPRT (**Gay et al., 2013**) alleles have been previously described. *Rosa26*$^{LSL-tdTomato}$ and CAG-LSL-HA-UPRT mice were obtained from the Jackson Laboratories (Bar Harbor, Maine). All animals were maintained on a mixed C57BL/6J × 129SvJ background. Mice were infected intratracheally with adenovirus (University of Iowa,

Gene Transfer Vector Core) or lentivirus as described (*DuPage et al., 2009*). Animal studies were approved by the University of Utah IACUC, and conducted in compliance with the Animal Welfare Act Regulations and other federal statutes relating to animals and experiments involving animals and adhere to the principles set forth in the Guide for the Care and Use of Laboratory Animals, National Research Council (PHS assurance registration number A-3031–01).

## Tamoxifen administration

Tamoxifen (Sigma, St. Louis, MO) was dissolved in corn oil to a concentration of 20 mg/ml and administered at a dose of 120 mg/kg per day for 6 doses over 9 days. This was followed by *ad libitum* feeding with tamoxifen-supplemented chow (500 mg/ kg; Envigo, Indianapolis, IN) in place of standard chow for the duration of experiment.

## Lentiviral production

Lentivirus was produced by transfection of 293 T cells with TransIT-293 (Mirus Bio, Madison, WI), lentiviral backbone as well as packaging vectors Δ8.9 (gag/pol) and CMV-VSV-G (*DuPage et al., 2009*). Supernatant was collected at 36, 48, 60 and 72 hr after transfection. For in vivo infection, virus was concentrated by ultracentrifugation at 25,000 r.p.m. for 105 min and re-suspended in an appropriate volume of 1X PBS. Cell line identity was authenticated using STR analysis at the University of Utah DNA Sequencing Core. Cells tested negative for mycoplasma.

## Cloning

We first generated a pCDH-SPC-Flpo lentiviral vector by PCR amplifying the murine SPC promoter (*Sutherland et al., 2011*) and cloning into SpeI-XbaI sites of pCDH-CMV-Flpo plasmid. The pCDH-mSPC-Flpo vector was then digested with ClaI-PacI and blunt ended with Klenow to clone into EcoRV site of the adenovirus shuttle plasmid G0687 pacAd5mcsSV40pA (University of Iowa, Viral Vector Core Facility). Correct identity and orientation of the construct was confirmed via Sanger sequencing. Further recombination and adenovirus production and purification was carried out by University of Iowa Viral Vector Core (cat.# VVC-Snyder-6695).

## Histology and immunohistochemistry

All tissues were fixed in 10% formalin overnight, and lungs were perfused with formalin via the trachea. Tissues were transferred to 70% ethanol, embedded in paraffin, and four-micrometer sections were cut. To detect mucin, sections were stained with 1% Alcian Blue pH 2.5 at the HCI Research Histology Shared Resource. Immunohistochemistry (IHC) was performed manually on Sequenza slide staining racks (ThermoFisher Scientific, Waltham, MA). Sections were treated with Bloxall (Vector labs) followed by Horse serum (Vector Labs, Burlingame, CA) or Rodent Block M (Biocare Medical, Pacheco, CA), primary antibody, and HRP-polymer-conjugated secondary antibody (anti-Rabbit, Goat and Rat from Vector Labs; anti-Mouse from Biocare. The slides were developed with Impact DAB or VIP (Vector) and counterstained with hematoxylin. Slides were stained with antibodies to BrdU (BU1/75, Abcam, Cambridge, MA), Cadherin 13 (EPR9621, Abcam), Cathepsin E (LS-B523, Lifespan Biosciences, Seattle, WA), Caveolin 1 (EPR15554, Abcam), CHIL3/4 (EPR15263, Abcam), Cleaved caspase-3 (5A13, CST, Danvers, MA), Cytokeratin 5 (EP1691Y, Abcam), Cytokeratin 7 (EP17078, Abcam), Cytokeratin-8 (TROMA-I, DSHB, Iowa City, Iowa), Cytokeratin 14 (EPR17350, Abcam), FoxA1 (EPR10881-14, Abcam), FoxA2 (EPR4466, Abcam), Galectin 4 (AF2128, R and D Systems, Minneapolis, MN), Gastrokine 1 (2E5, Abnova, Taipei City, Taiwan), GDA (EPR18751, Abcam), HNF4α (C11F12, CST), KI67 (SP6, Abcam), MCM2 (ab31159, Abcam), MMP7 (AF2967, R and D Systems) Muc5AC (SPM488, Abnova), NKX2-1 (EP1584Y, Abcam), p40(ΔNp63) (BC28, Biocare), PDX1 (F109-D12, DSHB), PIGR (7C1, Abcam), PK-LR (EPR11093P, Abcam), RFP (Rockland Immunochemicals, Limerick, PA), SOX2 (C70B1, CST) and VCAM1 (EPR5047, Abcam). Pictures were taken on a Nikon Eclipse Ni-U microscope with a DS-Ri2 camera and NIS-Elements software. For double immunostaining, slides were blocked sequentially with Bloxall, horse serum and Rodent Block M, then incubated with antibodies of interest from different species (Rabbit and Mouse) simultaneously. Slides were incubated with a mouse secondary followed by DAB (brown). This was followed by incubation with a rabbit secondary antibody and ImPACT VIP (purple, Vector lab). Tumor quantitation

was performed on hematoxylin and eosin-stained or IHC-stained slides using NIS-Elements software. All histopathologic analysis was performed by a board-certified anatomic pathologist (E.L.S.).

## Fluorescence-activated cell sorting (FACS)

7–20 weeks after tumor initiation with Ad5-SPC-Cre (*Sutherland et al., 2011*), tumor-bearing mice were euthanized using carbon dioxide and the rib-cage was dissected to reveal trachea and heart. Cadiac perfusion of the pulmonary vasculature was performed using PBS until the lungs turned pale. The lungs were inflated with an enzymatic digest solution (Collagenase type I (Thermo Fisher Scientific); Elastase (Worthington Biochemical, Lakewood, NJ), Dispase (Corning CB-40235, VWR, Radnor, PA) and Dnase I (DN25, Sigma)) and then minced and digested with the enzyme digest solution at 37 C for 45 min. The digested tissue was then passed through an 18-gauge syringe needle followed by 100, 70 and 40 micron filters to generate a single-cell suspension. The suspension was treated with Red Blood Cell Lysis Buffer (eBioscience, ThermoFisher Scientific,) and then reconstituted in 1X PBS supplemented with 2% fetal bovine serum, 2% BSA and DAPI (Sigma). Cells were sorted using BD FACSAria for tdTomato-positive and DAPI-negative cells into PBS + 10% serum.

## Single-cell isolation and RNA sequencing

Sorted tumor cells (200–300 cells/ul) were mixed with C1 Suspension Reagent (Fluidigm, South San Francisco, CA) and loaded on a 5–10 μm C1 Single-cell Auto Prep IFC for mRNA Seq (Fluidigm cat# 100–5760). Captured cells were visualized and scored by microscopy. Amplified cDNA products derived from captured cells were harvested and concentrations were measured using the Qubit dsDNA HS Assay Kit. Amplified products were normalized to a concentration of 0.2 ng/ul and sequencing libraries were prepared using the Nextera XT DNA Library Preparation Kit (cat# FC131-1096, Illumina, San Diego, CA) and dual indexed adapters (FC-131–2001, FC-131–2002) according to the modified protocol described by Fluidigm. Purified libraries were qualified on an Agilent Technologies 2200 TapeStation using a D1000 ScreenTape assay (cat# 5067–5582 and 5067–5583). The molarity of adapter-modified molecules was defined by quantitative PCR using the Kapa Biosystems (Wilmington, MA) Kapa Library Quant Kit (cat# KK4824). Individual libraries were normalized to 10 nM and equal volumes were pooled in preparation for Illumina sequence analysis.

Sequencing libraries (25 pM) were chemically denatured and applied to an Illumina HiSeq v4 single-single read flow cell using an Illumina cBot. Hybridized molecules were clonally amplified and annealed to sequencing primers with reagents from an Illumina HiSeq SR Cluster Kit v4-cBot (GD-401–4001). Following transfer of the flowcell to an Illumina HiSeq 2500 instrument (HCSv2.2.38 and RTA v1.18.61), a 50-cycle single-read sequence run was performed using HiSeq SBS Kit v4 sequencing reagents (FC-401–4002).

## Processing and analysis of single-cell RNA-seq data

### Transcript expression estimation

The genome index was created with STAR (v2.4.2a) (*Dobin et al., 2013*) using the mm10 genome sequence and Ensembl (build 85) gene definitions. Reads were aligned to the index using the following parameters: outFilterType BySJout, outFilterMultimapNmax 20, outFilterMismatchNmax 999, outFilterMismatchNoverReadLmax 0.04, alignIntronMin 20, alignIntronMax 1000000, alignMatesGapMax 1000000, alignSJoverhangMin 8, alignSJDBoverhangMin 1, sjdbScore 1, outSAMtype BAM SortedByCoordinate, quantMode TranscriptomeSAM.

The RSEM (v1.2.19) (*Li and Dewey, 2011*) reference was created using the rsem-prepare-reference command. Gene estimates were generated by running rsem-calculate-expression on the STAR alignments.

### Clustering

The gene count estimates from RSEM were loaded into a scater (v1.2.0) (*McCarthy et al., 2017*) SCESet object. Genes with log2(CPM) >2 in at least 10 cells were retained in the analysis. Cells with greater than 20% mitochondrial reads, less than 500 thousand alignments, less than 500 measurable genes or less than 20% mRNA bases were removed from the analysis. SC3 (v1.3.18) (*Kiselev et al., 2017*) was run on the filtered SCESet object using k = 3 and gene filtering turned off. The resulting

cell cluster assignments and marker genes were used in the remaining analyses. The scater 'plotTSNE' function was used to generate t-distributed stochastic neighbor embedding (t-SNE) plot.

AT2 gene count estimates (E18.5 cells, Treutlein et al.) were added to the passing cells from the previous analysis. SC3 and scater were run using the parameters listed above.

## Differential expression

Differential expression between the clusters was determined using the Bioconductor package SCDE (v1.99.1) (*Kharchenko et al., 2014*). RSEM gene count estimates from cells passing filtering were used in this analysis. Genes were retained if there were 10 or more counts in at least 10 cells. Error models were fit using the 'scde.error.models' function, expression magnitude priors for the genes were generated using the 'scde.expression.prior' function and differential expression was determined with the 'scde.expression.difference' function set to 100 randomizations.

## Correlation with bulk RNA-Seq data

The differential expression results from the bulk cell and single cell analyses were intersected. Genes with an average count less than 1000 in the bulk cell samples were removed. The log2 fold change values from both analyses were plotted using ggplot2 (v2.2.1) (*ggplot, 2009*). The Pearson correlation coefficient was calculated using the base R (v3.3.2) function cor.test.

## IMA signature

Transcripts per Million (TPM) estimates from RSEM were extracted for cells passing filtering and restricted to genes found in the human IMA signature (*Guo et al., 2017*). The FactoMineR (v1.39) (*Le et al., 2008*) function 'PCA' was run on the log-transformed TPM values and the first two components were plotted using ggplot.

## Normal tissue classification

TPM values were generated for each cluster by summing gene counts across the members of the cluster and dividing by the RSEM estimated gene length in kilobases to get the counts per base rate of each gene. These rates are divided by the sum of all rates and scaled by a million to get TPM. The TPM values for each cluster were then intersected with bulk cell TPM and normal tissue TPM downloaded from Encode (*Supplementary file 5*). Gene were restricted to those classified as 'protein coding' by Ensembl and had at least 10 counts in 10 or more cells.

High-expressing genes for each normal tissue were selected by first calculating the average tissue log2 CPM for each gene, which were then mean-centered across all tissues. The tissue with the highest expression was assigned its own mean-centered expression value. Once all genes were processed, the assigned genes in each tissue were ranked by expression and the top 70 were reported.

The Rtnse (v0.13) (*Krijthe, 2018*) function 'Rtsne' was used to generate a tSNE plot on the log2 TPM values of the tissue enriched genes. The perplexity was set to 13 and the initial dimensions was set to 5.

The FactoMineR function 'PCA' was run on the log2 TPM values with scaling turned off and five dimensions. The FactorMineR function 'HCPC' was then used perform a hierarchical clustering on principle component (HCPC) analysis on the PCA result.

Cosine similarity was calculated using the lsa (v0.73.1) (*Wild, 2015*) function 'cosine'. Every combination of cluster and normal tissue log2 TPM values were compared.

## Bulk RNA isolation and total RNA-Seq

RNA was isolated by trizol-chloroform extraction followed by column-based purification. Sorted cells were lysed in 1 ml Trizol (ThermoFisher Scientific), followed by phenol-chloroform extraction. The aqueous phase was brought to a final concentration of 50% ethanol, and RNA was purified using the PureLink RNA Mini kit according to the manufacturer's instructions (ThermoFisher Scientific). Library preparation was performed using the TruSeq Stranded RNA kit with Ribo-Zero Gold (Illumina). Libraries were sequenced on an Illumina HiSeq 2500 (50 cycle single-read sequencing).

## Processing and analysis of total RNA-seq data

Mouse FASTA and GTF files were downloaded from Ensembl release 82 and a reference database was created using RSEM version 1.2.12 (*Li and Dewey, 2011*). RSEM and the Bowtie 1.0.1 aligner were used to map reads and estimate transcripts and gene counts using rsem-calculate-expression with the forward-prob 0 option for reversely stranded Illumina reads. The expected gene counts were filtered to remove 12371 features with zero counts and 10100 features with fewer than 10 reads in any sample. Differentially expressed genes were identified using a 5% false discovery rate with DESeq2 version 1.16.0 (59).

## Histopathologic evaluation of primary human tumors

Formalin fixed, paraffin-embedded (FFPE) tumors were obtained in accordance with protocols approved by the Institutional Review Boards of the University of Utah and Intermountain Healthcare. Additional lung adenocarcinomas were evaluated on commercially available tissue microarrays (US BioMax, Rockville, MD).

## Comparison of *FOXA1* and *FOXA2* levels in *KRAS*-mutant human lung adenocarcinomas

The patient IDs and cluster names from 68 KRAS-mutants listed in supplementary figure 2A in *Skoulidis et al. (2015)* were saved to a sample table with 23 KL, 30 KP and 15 KC samples corresponding to genetic alterations in STK11/LKB1 (KL), TP53 (KP), and CDKN2A/B inactivation coupled with low expression of NKX2-1 (KC). The patient IDs were matched to a count matrix from the TCGA Lung Adenocarcinoma project (LUAD) using the TCGAbiolinks package and HTSeq counts in the GDC harmonized dataset (*Colaprico et al., 2016*). Eleven patients with a matched normal sample were also included as a fourth group for comparison. The count matrix was filtered to remove 5789 features with zero counts and 19,546 features with fewer than 10 reads in any sample. The sample table and filtered count matrix were loaded into DESeq2 version 1.16.0 (59) to estimate normalized counts and identify differentially expressed genes using a 5% false discovery rate.

## Statistics

*p*-Values were calculated using the unpaired two-tailed Mann-Whitney (non-parametric) U test, Chi-squared test or Fisher's Exact Test. RNA-Seq statistics are described above.

# Acknowledgements

We are grateful to members of the Snyder lab for suggestions and comments. We thank Brian Dalley for sequencing expertise and James Marvin for FACS expertise. Core facilities (BMP, Genomics/Bioinformatics, Flow Cytometry). Research reported in this publication utilized shared resources (including Flow Cytometry, High Throughput Genomics, Bioinformatics, and Biorepository and Molecular Pathology) at the University of Utah and was supported by the National Cancer Institute of the National Institutes of Health under Award Number P30CA042014. Work in the flow cytometry core was also supported by the National Center for Research Resources of the National Institutes of Health under Award Number 1S20RR026802-1. ELS was supported in part by a Career Award for Medical Scientists from the Burroughs Wellcome Fund, a V Scholar Award, the NIH (R01CA212415) and institutional funds (Department of Pathology and Huntsman Cancer Institute, University of Utah).

# Additional information

### Funding

| Funder | Grant reference number | Author |
| --- | --- | --- |
| National Cancer Institute | R01212415 | Eric L Snyder |
| Burroughs Wellcome Fund | Career Award for Medical Scientists | Eric Snyder |
| V Foundation for Cancer Research | Scholar Award | Eric Snyder |

The funders had no role in study design, data collection and interpretation, or the decision to submit the work for publication.

## Author contributions
Soledad A Camolotto, Grace Orstad, Investigation, Writing—original draft, Writing—review and editing; Shrivatsav Pattabiraman, Investigation, Writing—original draft; Timothy L Mosbruger, Formal analysis, Writing—original draft, Writing—review and editing; Alex Jones, Investigation, Writing—review and editing; Veronika K Belova, Mitchell Streiff, Lydia Salmond, Investigation; Chris Stubben, Formal analysis; Klaus H Kaestner, Resources, Writing—review and editing; Eric L Snyder, Conceptualization, Resources, Formal analysis, Supervision, Funding acquisition, Investigation, Visualization, Methodology, Writing—original draft, Project administration, Writing—review and editing

## Author ORCIDs
Klaus H Kaestner (iD) http://orcid.org/0000-0002-1228-021X
Eric L Snyder (iD) http://orcid.org/0000-0003-3591-3195

## Ethics
Animal experimentation: This study was performed in strict accordance with the recommendations in the Guide for the Care and Use of Laboratory Animals of the National Institutes of Health. All of the animals were handled according to approved institutional animal care and use committee (IACUC) protocols (#15-07009) of the University of Utah.

## Decision letter and Author response
Decision letter https://doi.org/10.7554/eLife.38579.026
Author response https://doi.org/10.7554/eLife.38579.027

# Additional files

## Supplementary files
• Supplementary file 1. List of all single cells used in analysis with quality control metrics, genotype of mouse and cluster each cell was assigned. In cells that failed quality control, cluster is not assigned and failed QC metric(s) are highlighted.
DOI: https://doi.org/10.7554/eLife.38579.016

• Supplementary file 2. Table of gene expression levels in all high quality cells used in downstream analysis. Cells are organized by cluster (C1: gray, C2: yellow, C3: green).
DOI: https://doi.org/10.7554/eLife.38579.017

• Supplementary file 3. Marker genes from each cluster (generated with S3). Genes differentially expressed between each cluster (pairwise comparisons, SCDE).
DOI: https://doi.org/10.7554/eLife.38579.018

• Supplementary file 4. Genes differentially expressed between tumor cells sorted from $Kras^{LSL-G12D/+}$ mice (**K**, n = 3 mice) and $Kras^{LSL-G12D/+}$; $Nkx2-1^{F/F}$ mice (**KN**, n = 3 mice)
DOI: https://doi.org/10.7554/eLife.38579.019

• Supplementary file 5. List of normal murine tissues used and their source.
DOI: https://doi.org/10.7554/eLife.38579.020

• Supplementary file 6. Cosine similarity table quantitating similarity between single cell clusters and each normal tissue evaluated.
DOI: https://doi.org/10.7554/eLife.38579.021

• Transparent reporting form
DOI: https://doi.org/10.7554/eLife.38579.022

## Data availability
All data generated or analysed during this study are included in the manuscript and supporting files. Sequencing data will be deposited in GEO under accession codes GSE115901.

The following dataset was generated:

| Author(s) | Year | Dataset title | Dataset URL | Database and Identifier |
| --- | --- | --- | --- | --- |
| Snyder E | 2018 | FoxA1 and FoxA2 are required for gastric differentiation in NKX2-1-negative lung adenocarcinoma | https://www.ncbi.nlm.nih.gov/geo/query/acc.cgi?acc=GSE115901 | NCBI Gene Expression Omnibus, GSE115901 |

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
