## [Decision Letter]

Thank you for submitting your manuscript entitled "FoxA1 and FoxA2 are required for gastric differentiation in NKX2-1-negative lung adenocarcinoma" for peer review at *eLife*. Your article has been evaluated by three peer reviewers, and the evaluation was overseen by a Reviewing Editor and a Senior Editor.

As you will see below, each of the reviewers was generally enthusiastic about the fundamental observations you made about FoxA genes in lung cancer. However, overall, there was significant concern that you have not really been able to discern two potential explanations for your findings: 1) a single cell of origin accompanied by an epigenetic shift, versus 2) two different cells of origin. This is considered a major impediment to publication since these would suggest significantly different mechanisms for your observations.

We recognize that clarifying this experimentally may be very challenging, especially within the two month time frame for eLife revisions. Based on this, what I would request is that you look over the critiques below and provide a detailed plan for what you think you could achieve to address them within a two month window. We will then look over your proposed plan of action and editorially decide if we want to proceed. If we decide to proceed, we would then give you two months to accomplish what you stated and have the revised manuscript re-reviewed by the original reviewers.

*Reviewer #1:*

In this study, Camolotto et al. have investigated lung adenocarcinoma progression with a specific focus on the mechanisms of how cellular identity evolves in Nkx2.1 mutant invasive mucinous adenocarcinomas. They conclude that both *Foxa1* and *Foxa2* are required for the maintenance of the adenocarcinoma phenotype. Interestingly, the timing of *Foxa1/2* inactivation determined the outcome of the eventual tumor phenotype.

The study is carefully executed and employs state-of-the-art approaches such a series of elegant genetically engineered mouse models as well as single-cell mRNA sequencing. The manuscript is well-written and the data are clearly presented. The manuscript is important, providing mechanistic insight into how cellular identity is transcriptionally regulated. The manuscript motivates further studies into identifying how epigenetic and genetic states dictate the outcome of lineage-determining transcription factors.

Major concern

The genetic evidence for the role of *Foxa1/2* in preserving lung adenocarcinoma identity in the mouse model is clear. However, the manuscript would greatly benefit from a comparison with human adenosquamous lung cancers. In human IMA, Nkx2.1 is often inactivated through point mutations that do not necessarily inactivate all functions of the protein (DNA binding, *Foxa1/2* binding, association with upstream regulators). In contrast, the mouse model produces a full deletion of both copies of Nkx2-1, leading to complete absence of the protein. Thus, the mechanism by which adenosquamous tumors arise may be different. At least computational analyses or additional discussion should be provided.

Overall, this is a high quality study and appropriate for the readership of *eLife*.

*Reviewer #2:*

In this manuscript Camolotto et al. provide evidence for *Foxa1* and *Foxa2* activating the expression of gastrointestinal genes in *Kras^LSL-G12D/+^*;*Nkx2-1^F/F^* driven ADC lesions in mice carrying either *Foxa1/2* deletions. It has been previously described that normal and neoplastic epithelial cells adopt a gastric differentiation state after *Nkx2-1* deletion and that the re-localization of the transcription factors *Foxa1* and *Foxa2* from pulmonary to gastrointestinal genes is the mechanism responsible for this change in differentiation (Snyder et al., 2013).

The authors show that *Foxa1/2* deletions correlates, in some cases, with squamous differentiation of tumor lesions. Since lung SCC is found adjacent to lung ADC (Adenosquamous lesions) in mice genetically proficient for either *Foxa1* or *Foxa2*, but actually negative for the expression of both, they suggest that only when the expression of one of the two genes is stochastically lost, mice develop SCC. In contrast, when both *Foxa1* and *Foxa2* are genetically deleted from the beginning, mice develop ADC expressing markers of the squamo-columnar junction of the gastrointestinal tract. They suggest that this difference is attributable to a different context in which the lineage specifier is lost.

In general, the idea that the effects of lineage specifier inactivation in cancer can be highly context-dependent is intriguing. However, the experiments as described lack important controls and quantifications to firmly conclude this.

Point by point discussion:

In invasive mucinous ADC (IMA) (Guo et al., 2017) *Foxa2* is not found commonly upregulated. However, *Foxa3* is, and it produces mucinous ADC in transgenic mouse models. Actually, *Foxa3* and 1 are the commonly upregulated genes in human and mouse IMA (Guo M). Does the expression of *Foxa3* change in mice upon *Foxa1/2* deletion? Can the authors exclude that HNF4α expression in *Foxa1/2^F/F^*mice is activated by *Foxa3*? To confirm that *Foxa1/2* are indeed re-localized upon *Nkx2-1* deletion, include *Kras^LSL-G12D/+^* mice proficient for *Nkx2-1* in the IHC panel of Figure 1. Add *Foxa3* staining.

The authors report a "variable but substantial" quantity of "incomplete recombinants" when they refer to mice carrying the double conditional deletion of *Foxa1* and *Foxa2*. This needs substantiation at the different time points. If incomplete recombination is responsible for the tumor outgrowth one would expect regions that are highly positive for either FoxA1 or A2 (as seen in the *Foxa1* or *Foxa2* single knockouts). However, the staining is rather dispersed as if deletion of non-recombined alleles continues. How do the authors reconcile this with the prolific outgrowth of partial recombinants? If recombination does indeed proceed (through lentiviral Cre expression) this then should promote squamous differentiation according to the model the authors propose but this does not occur.

Can the authors formally exclude that instead of incomplete recombination, the result is due to the fact that *Foxa1* and *Foxa2* deletion is just not sufficient to suppress the gastric differentiation? What about FoxA3?

The authors use subsequently 2 different models to support the context-dependence of LCC development: i) *Kras^FSF-G12D/+^*;*Rosa^FSF-CreERT2^; Nkx2-1^F/F^*; *Foxa1F/F; FoxaF/F* mice in which mutant *Kras* is first activated via Ad5CMV-FlpO, and one week later, tamoxifen is administered to activate *Foxa1/2* deletions. The sequential introduction of these lesions results in LSCC which is already detectable at 5 weeks. ii) *KrasG12D /+; Nkx2-1F/F*; *Foxa1F/F; FoxaF/F* mice infected with Ad5-CMV-Cre; at 5 weeks no LSCC is found. They conclude that in the context in which mutant *Kras* is already activated, *Foxa1/2* deletion gives rise to SCC; when mutations occur at the same time, mice develop ADC. This possible, but this requires better control of the experiment. They need to assess potential effects of tamoxifen treatment (e.g. by giving Tamoxifen also in the model in "ii". They should also analyze mice after a longer period.

Can the authors exclude that markers for the squamo-columnar junction cells of the gastrointestinal tract, are shared with precursors of lung SCC? It would help to include other markers of SCC, such as K5, K14, SOX2, which are also robust markers and are already expressed in intermediate lesions committed to squamous differentiation.

*Reviewer #3:*

Camolotta and colleagues examine the consequences of deleting *Nkx2-1, Foxa1* and *Foxa2* in the *Kras^G12D^* lung adenocarcinoma model. Previously, they had shown that deletion of *Nkx2-1* in *Kras^G12D^* -driven lung adenocarcinoma results in mucinous conversion with corresponding molecular switch from lung to gastric epithelial identity. Here, they report that deletion of *Nkx2-1, Foxa1* and *Foxa2* following *Kras^G12D^* initiation results in squamous transdifferentiation with lack of gastric marker expression. However, when the factors are deleted concomitant with *Kras^G12D^* induction, a different tumor phenotype results and cells have molecular features of the squamocolumnar junction (SCJ) of the GI tract. The authors speculate that the differential effect from sequential manipulation results from epigenetic changes induced by Kras activity.

The main conclusion of the work and title of the manuscript, that it is *Foxa1* and *Foxa2* that mediate the gastric differentiation in *Nkx2-1* negative lung adenocarcinoma, is entirely unsurprising. *Foxa1* and *Foxa2* are already known to directly regulate expression of gastric target genes, so clearly their deletion in *Nkx2-1* deficient tumor cells should compromise expression of gastric markers. This finding by itself does not really represent a significant advance in the field.

The different phenotypes resulting from concurrent deletion and activation of *Kras^G12D^* versus deletion seven days after induction of *Kras* is very interesting, but no evidence that this is due to epigenetic changes induced by *Kras* is provided. It seems at least equally plausible that the distinct tumor histology represents transformation of a different (non-AT2) cell type. This alternate model would also explain why there was a reduction in tumor number, since if only the molecular identity and proliferation of recombined AT2 cells were influenced, one might expect the same number of tumors but of smaller size. There were also fewer tumors when only *Foxa1* was deleted, which they did not attempt to explain. Unfortunately, without inducing *Kras^G12D^* in a cell type-specific manner, it is impossible to deconvolute the cell of origin retrospectively, particularly when the tumor lacks expression of lung cell type markers. So, I don't feel there is really any reasonably supported explanation for this important finding, and whether it results from an epigenetic effect versus a different cell of origin would drastically alter the model.

Regarding the possibility that the molecular phenotype of these small tumors is of SCJ cells, it is unclear why this would be the case, since SCJ cells apparently express *Foxa1*. Wouldn't deletion of *Nkx2-1* and *Foxa2* thus be more likely to induce a SCJ identity than deletion of all three factors? So, without further exploration this interpretation seems provisional.

---

## [Author Response]

[Editors' note: the authors’ plan for revisions was approved and the authors made a formal revised submission.]

As you will see below, each of the reviewers was generally enthusiastic about the fundamental observations you made about FoxA genes in lung cancer. However, overall, there was significant concern that you have not really been able to discern two potential explanations for your findings: 1) a single cell of origin accompanied by an epigenetic shift, versus 2) two different cells of origin. This is considered a major impediment to publication since these would suggest significantly different mechanisms for your observations.

We are grateful to the reviewers for their thoughtful and constructive comments, and appreciate their overall enthusiasm for our work. We are excited to report that we have been able to develop a novel adenovirus that allows us to address the major concern about cell of origin enumerated in the summary letter. In brief, we now show that SPC-positive lung epithelial cells can give rise to squamous cell carcinoma upon loss of NKX2-1, FoxA1 and FoxA2. Please see below for detailed response to each reviewer.

Reviewer #1:[…] The genetic evidence for the role of Foxa1/2 in preserving lung adenocarcinoma identity in the mouse model is clear. However, the manuscript would greatly benefit from a comparison with human adenosquamous lung cancers. In human IMA, Nkx2.1 is often inactivated through point mutations that do not necessarily inactivate all functions of the protein (DNA binding, Foxa1/2 binding, association with upstream regulators). In contrast, the mouse model produces a full deletion of both copies of Nkx2-1, leading to complete absence of the protein. Thus, the mechanism by which adenosquamous tumors arise may be different. At least computational analyses or additional discussion should be provided.

We agree that our murine studies raise the question of whether loss of FoxA1/2 expression is also observed in human AdSCC. To address this question, we have obtained tissue sections from 12 primary human AdSCC cases in which levels of both FoxA1 and FoxA2 could be evaluated by IHC in both the adenocarcinoma (ACA) and squamous (SCC) components. In six cases, FoxA1 and FoxA2 were both downregulated in the SCC component compared with the ACA component. In five cases, only FoxA2 was downregulated in the SCC (FoxA1 was expressed in both components) and in one case, FoxA1 was downregulated in SCC (FoxA2 was expressed in both components). We have added a graph and representative images (Figure 4D-E) reflecting these new data. These data show that FoxA1/2 downregulation is associated with squamous differentiation in human AdSCC. Human cancers are more heterogeneous than any given mouse model, so it is not surprising that we observe more than one distinct pattern of changes in FoxA1/2 levels in these cases. Nevertheless, the fact that all cases exhibited lower levels of FoxA1 and/or FoxA2 in the SCC component suggests that our results are highly relevant to the human disease, and that loss of these transcription factors may also be causative for adenosquamous transdifferentiation in human lung cancer. Although we don’t have access to genetic analysis of these samples, we predict that mutational spectrum and/or cell of origin may dictate whether loss of one or both FoxA transcription factors is sufficient for squamous transdifferentiation in this disease.

Overall, this is a high quality study and appropriate for the readership of eLife.Reviewer #2:[…] In general, the idea that the effects of lineage specifier inactivation in cancer can be highly context-dependent is intriguing. However, the experiments as described lack important controls and quantifications to firmly conclude this.Point by point discussion:

*In invasive mucinous ADC (IMA) (Guo et al., 2017) Foxa2 is not found commonly upregulated. However, Foxa3 is, and it produces mucinous ADC in transgenic mouse models. Actually, Foxa3 and 1 are the commonly upregulated genes in human and mouse IMA (Guo M). Does the expression of Foxa3 change in mice upon Foxa1/2 deletion? Can the authors exclude that HNF4α expression in Foxa1/2^F/F^mice is activated by Foxa3? To confirm that Foxa1/2 are indeed re-localized upon Nkx2-1 deletion, include KrasLSL-G12D/+ mice proficient for Nkx2-1 in the IHC panel of Figure 1. Add Foxa3 staining.*

Although *Foxa2* is not commonly upregulated at the mRNA level in human IMA, it is readily detectable at the protein level in this disease (Figure 1—figure supplement 1A). Our single cell RNA-Seq data (Supplementary file 3) shows that *Foxa3* is more highly expressed in cluster C2 (KN) than C3 (KNF1F2) or C1 (K), demonstrating that FoxA1/2 drive *Foxa3* expression in NKX2-1-negative tumors. (*Foxa3* is also higher in KN vs. K tumors in our total RNA-Seq analysis, Supplementary file 4). Unfortunately, we were not able to identify commercially available IHC-quality antibody to characterize FoxA3 at the protein level. Review of the literature indicates that the polyclonal antibody sc-5361 has been used by other authors in IHC. However, this antibody has been discontinued by the manufacturer (Santa Cruz). Nevertheless, it is clear from our RNA-Seq data that FoxA1/2 are required for expression of both HNF4α and FoxA3. The experiments in this manuscript are not designed to test the regulatory interaction between HNF4α and FoxA3. There are several lineage specifiers regulated by FoxA1/2 in NKX2-1-negative lung adenocarcinoma (including PDX1, HNF4γ and HNF1α), and in future work it would be interesting to define their hierarchical regulatory interactions. We do not believe it is necessary to include IHC on NKX2-1 positive tumors because they have been shown to lack gastric markers in Snyder et al., 2013 and Maeda et al. JCI 2012. This is further demonstrated by the RNA-seq data presented in this manuscript (Supplementary files 3-4). We also extensively characterized FoxA1/2 re-localization by ChIP-seq in Snyder et al., 2013.

The authors report a "variable but substantial" quantity of "incomplete recombinants" when they refer to mice carrying the double conditional deletion of Foxa1 and Foxa2. This needs substantiation at the different time points. If incomplete recombination is responsible for the tumor outgrowth one would expect regions that are highly positive for either FoxA1 or A2 (as seen in the Foxa1 or Foxa2 single knockouts). However, the staining is rather dispersed as if deletion of non-recombined alleles continues. How do the authors reconcile this with the prolific outgrowth of partial recombinants? If recombination does indeed proceed (through lentiviral Cre expression) this then should promote squamous differentiation according to the model the authors propose but this does not occur.Can the authors formally exclude that instead of incomplete recombination, the result is due to the fact that Foxa1 and Foxa2 deletion is just not sufficient to suppress the gastric differentiation? What about FoxA3?

In the original manuscript, we carefully quantitated the relative proportion of incomplete recombinants at multiple timepoints in Figure 2—figure supplement 1A. In this graph, we show that the proportion of HNF4α-positive incomplete recombinants increases over time, consistent with the growth advantage we would expect from the data presented in Figure 2. These HNF4α-positive incomplete recombinants always retain either FoxA1 and/or FoxA2 expression. Thus, we infer that expression of Cre has been lost in these lesions. This is an expected result since lentiviral integration into the genome is random and genes delivered by the lentivirus can be readily downregulated. We have added a photomicrograph of an incomplete recombinant to further illustrate this point (Figure 2—figure supplement 1B).

In response to the comment: “the staining is rather dispersed as if deletion of non-recombined alleles continues”.If thisrefers to Figure 1, we want to clarify that the photomicrographs in this figure all show complete recombinants. FoxA1/2-positive cells in pictures from *Foxa1/2* conditional mice are normal type 2 pneumocytes that are dispersed throughout the alveoli and sometimes found intermingled with neoplastic lesions of all genotypes. To be clear, complete recombinants in *Foxa1/2* conditional mice exhibit no staining for FoxA1 or FoxA2 protein by IHC, and are morphologically quite distinct from incomplete mucinous recombinants.

Overall, our data are conclusive that *Foxa1/2* deletion is sufficient to suppress gastric differentiation, including *Foxa3* expression.

The authors use subsequently 2 different models to support the context-dependence of LCC development: i) KrasFSF-G12D/+; RosaFSF-CreERT2; Nkx2-1F/F; Foxa1F/F; FoxaF/F mice in which mutant Kras is first activated via Ad5CMV-FlpO, and one week later, tamoxifen is administered to activate Foxa1/2 deletions. The sequential introduction of these lesions results in LSCC which is already detectable at 5 weeks. Ii) KrasG12D /+; Nkx2-1F/F; Foxa1F/F; FoxaF/F mice infected with Ad5-CMV-Cre; at 5 weeks no LSCC is found. They conclude that in the context in which mutant Kras is already activated, Foxa1/2 deletion gives rise to SCC; when mutations occur at the same time, mice develop ADC. This possible, but this requires better control of the experiment. They need to assess potential effects of tamoxifen treatment (e.g. by giving Tamoxifen also in the model in "ii".

The mice that developed AdSCC in Figure 4 did not receive tamoxifen treatment, supporting the notion that tamoxifen is not likely to mediate or be required for squamous transdifferentiation. In addition, the experiments shown in Figure 5 include two controls of distinct genotypes (*Foxa1/2*^+/+^, *Foxa2*^F/F^ and *Foxa1/2*^F/F^). All three genotypes received tamoxifen, but only the latter group developed SCC. To further control for the effects of tamoxifen, we have included new data (Figure 6) showing that tamoxifen treatment of *Kras^LSL-G12D/+^; Nkx2-1^F/F^; Foxa1^F/F^; Foxa2^F/F^*mice infected with Ad5-SPC-Cre does not lead to SCC formation, whereas tamoxifen treatment of *Kras^FSF-G12D/+^;Rosa^FSF-CreERT2^; Nkx2-1^F/F^; Foxa1^F/F^; Foxa2^F/F^*mice infected with Ad5-SPC-FlpO does lead to SCC formation.

They should also analyze mice after a longer period.

We are currently aging a cohort of *Kras^FSF-G12D/+^; Rosa^FSF-CreERT2^; Nkx2-1^F/F^; Foxa1^F/F^; Foxa2^F/F^*mice (and of *Kras^FSF-G12D/+^; Rosa^FSF-CreERT2^; Nkx2-1^F/F^* controls) infected with low dose Ad5CMV-FlpO followed by tamoxifen 1 week later. This experiment is not complete, so we would prefer not to include it in the main manuscript. However, we can report that all four of the *Foxa1^F/F^; Foxa2^F/F^*mice analyzed by histopathology so far harbor multiple SCCs (some macroscopic in size) as well as microscopic SCJ-like lesions. In contrast, none of the five controls analyzed so far harbor SCCs, but rather contain mucinous lesions as expected from *Nkx2-1* deletion.

Although this experiment is informative and expands on the shorter term studies included in the manuscript, it does not alter any of our fundamental conclusions.

Can the authors exclude that markers for the squamo-columnar junction cells of the gastrointestinal tract, are shared with precursors of lung SCC?

As shown in Figure 3, markers of the SCJ are not expressed in invasive mucinous adenocarcinoma, which appears to be the precursor of the SCC component of the AdSCC lesions we describe in Figure 4. We have not identified a specific precursor lesion of the squamous lesions shown in Figures 5-6, so we can’t address that question for these lesions. It is formally possible that the squamous lesions shown in Figure 5-6 arise from a transient precursor expressing SCJ markers. Alternatively, these squamous lesions may arise directly from KRAS^G12D^-expressing cells upon deletion of *Nkx2-1, Foxa1* and *Foxa2*.

It would help to include other markers of SCC, such as K5, K14, SOX2, which are also robust markers and are already expressed in intermediate lesions committed to squamous differentiation.

IHC for SOX2 and K14 is shown in Figure 4—figure supplement 1C. Interestingly, SOX2 is not expressed in either the adenocarcinoma or SCC component of the tumors. K14 is upregulated in SCC. We have provided additional IHC for K5, which is positive in SCC as expected (Figure 3—figure supplement 1, Figure 4—figure supplement 1, and Figure 6).

Reviewer #3 (General assessment and major comments (Required)): [...] The different phenotypes resulting from concurrent deletion and activation of KrasG12D versus deletion seven days after induction of Kras is very interesting, but no evidence that this is due to epigenetic changes induced by Kras is provided. It seems at least equally plausible that the distinct tumor histology represents transformation of a different (non-AT2) cell type. This alternate model would also explain why there was a reduction in tumor number, since if only the molecular identity and proliferation of recombined AT2 cells were influenced, one might expect the same number of tumors but of smaller size. There were also fewer tumors when only Foxa1 was deleted, which they did not attempt to explain. Unfortunately, without inducing KrasG12D in a cell type-specific manner, it is impossible to deconvolute the cell of origin retrospectively, particularly when the tumor lacks expression of lung cell type markers. So, I don't feel there is really any reasonably supported explanation for this important finding, and whether it results from an epigenetic effect versus a different cell of origin would drastically alter the model.

To address the cell of origin question, we have generated an adenovirus expressing FlpO from the SPC promoter (Ad-SPC-FlpO). The SPC promoter has previously been used by Anton Berns and others to drive Cre expression specifically in SPC-positive type 2 pneumocytes of the lung (Sutherland et al., 2011). We used this virus to specifically target SPC-positive cells in *Kras^FSF-G12D/+^; Rosa^FSF-CreERT2^; Nkx2-1^F/F^; Foxa1^F/F^; Foxa2^F/F^*mice. As shown in the new Figure 6, infection with Ad-SPC-FlpO followed by tamoxifen 1 week later leads to the development of both keratinizing squamous cell carcinoma lesions and SCJ-like lesions at the 5 week timepoint (similar to the phenotypes we observed with the CMV-FlpO adenovirus, Figure 5). Thus, SCC can arise from SPC-positive cells. The fact that SPC-positive cells can frequently give rise to SCC in this experiment is particularly surprising given recent work (Nagaraj et al., 2017) showing that squamous lesions in *Kras; Lkb1* conditional mice arise almost exclusively from CC10-positive cells rather than SPC-positive cells. These data suggest an unexpected degree of lineage plasticity within the SPC-positive population.

Although these data show that loss of three lineage specifiers (NKX2-1, FoxA1 and FoxA2) is sufficient for SCC formation in SPC-positive cells, they also raise the question of why SPC-positive cells can give rise to two distinct phenotypes upon the same genetic manipulation (CK5-positive SCC vs. CK7-positive SCJ-like lesions). In the Discussion, we point to various potential sources of this heterogeneity, including cell cycle status at the time of lineage specifier deletion, heterogeneity within the SPC-positive population, or even purely stochastic elements. Investigating these alternatives will likely form the basis of a future manuscript.

Regarding the possibility that the molecular phenotype of these small tumors is of SCJ cells, it is unclear why this would be the case, since SCJ cells apparently express Foxa1. Wouldn't deletion of Nkx2-1 and Foxa2 thus be more likely to induce a SCJ identity than deletion of all three factors? So, without further exploration this interpretation seems provisional.

We agree that cancer cells are fundamentally sui generis and have tried not to overinterpret the data or argue that they are exactly the same as any normal cell type. Nevertheless, it is striking that many SCJ markers are upregulated in cancer cells that concomitantly lose NKX2-1, FoxA1 and FoxA2. Moreover, these tumor cells still cluster with the normal upper GI tract in our bioinformatic analysis. Taken together, these data suggest that they are more like SCJ cells than any other discrete cell type we have been able to identify. We respectfully disagree with the argument that deletion of *Nkx2-1* and *Foxa2* would be more likely to induce an SCJ-like identity. Through careful titration of FoxA1 IHC, we have found that FoxA1 levels at the SCJ are extremely low (much lower than FoxA1 levels in *Nkx2-1/Foxa2* deleted tumors) (Figure 3—figure supplement 1F vs. Figure 1). We have added arrows to this figure to make this point clear. Thus, we would argue that overall FoxA1/2 activity at the SCJ is likely to be much lower than in *Nkx2-1/Foxa2* deleted tumors. Unfortunately, there is no publicly available gene expression data for the recently described KRT7-positive SCJ cells identified by Jiang et al. (Jiang et al., 2017), and we do not have the tools to isolate these cells ourselves.

“There were also fewer tumors when only Foxa1 was deleted, which they did not attempt to explain.”

Although *Foxa1* deletion had a mild effect on tumor burden, it had no effect on proliferation rate or overall survival. Indeed, the most striking effect of *Foxa1* deletion was enhanced mucin secretion at late timepoints (Figure 2—figure supplement 1I). Although we suspect that FoxA1 and FoxA2 have some non-redundant functions in NKX2-1-negative tumors, we found that deleting both led to much more significant phenotypes, and thus we prioritized the analysis of complete FoxA1/2 loss as being the most impactful area of investigation.